# Newswire: A Large-Scale Structured Database of a Century of Historical News

**Emily Silcock**
Harvard University
Cambridge, MA 02478
emilysilcock@fas.harvard.edu

**Abhishek Arora**
Harvard University
Cambridge, MA 02478
abhishekarora@fas.harvard.edu

**Luca D'Amico-Wong**
Harvard University
Cambridge, MA 02478
ldamicowong@college.harvard.edu

**Melissa Dell**
Harvard University
Cambridge, MA 02478
melissadell@fas.harvard.edu

## Abstract

In the U.S. historically, local newspapers drew their content largely from newswires like the Associated Press. Historians argue that newswires played a pivotal role in creating a national identity and shared understanding of the world, but there is no comprehensive archive of the content sent over newswires. We reconstruct such an archive by applying a customized deep learning pipeline to hundreds of terabytes of raw image scans from thousands of local newspapers. The resulting dataset contains 2.7 million unique public domain U.S. newswire articles, written between 1878 and 1977. Locations in these articles are georeferenced, topics are tagged using customized neural topic classification, named entities are recognized, and individuals are disambiguated to Wikipedia using a novel entity disambiguation model. To construct the `Newswire` dataset, we first recognize newspaper layouts and transcribe around 138 million structured article texts from raw image scans. We then use a customized neural bi-encoder model to de-duplicate reproduced articles, in the presence of considerable abridgement and noise, quantifying how widely each article was reproduced. A text classifier is used to ensure that we only include newswire articles, which historically are in the public domain. The structured data that accompany the texts provide rich information about the who (disambiguated individuals), what (topics), and where (georeferencing) of the news that millions of Americans read over the course of a century. We also include Library of Congress metadata information about the newspapers that ran the articles on their front pages. The `Newswire` dataset is useful both for large language modeling - expanding training data beyond what is available from modern web texts - and for studying a diversity of questions in computational linguistics, social science, and the digital humanities.

## 1 Introduction

As large language models become more widely-used, researchers are likely to turn increasingly to the past in an effort to expand the world knowledge that they can access. The past is inherently of interest, both to researchers and the general public, and making past texts accessible to language models can increase our capacity to draw insights from history. While a much larger portion of historical data has entered the public domain compared to present-day data, historical texts are not as readily accessible as web texts. They often require sophisticated pipelines to extract before they can power downstream

38th Conference on Neural Information Processing Systems (NeurIPS 2024) Track on Datasets and Benchmarks.

applications, ranging from training large language models to conducting research in social science, computational linguistics, and the digital humanities.

News forms a central repository of past world knowledge. Because maintaining a global network to collect the news was expensive, newswires such as the Associated Press and United Press were a main source of news in the United States historically. Media historian Julia Guarneri (11) writes: "by the 1910s and 1920s, most of the articles that Americans read in their local papers had either been bought or sold on the national news market... This constructed a broadly understood American 'way of life' that would become a touchstone of U.S. domestic politics and international relations throughout the twentieth century." Despite its potential utility as a source of training data and its relevance to understanding the historical path that has shaped the present, there is no comprehensive dataset of the millions of texts sent out over newswires in the 19th and 20th centuries.

This is most likely because no comprehensive archive of these texts exists. Extant archives focus on single regional bureaus of a single wire service for limited dates. To the extent a digitization of archival materials exists, optical character recognition (OCR) errors are rampant. For instance, we found in the Associated Press Corporate Archives (Gale Primary Sources) that OCR typically transcribed over half of characters wrong, turning many into undecipherable punctuation or non-Latin characters.

Hence, we instead reconstruct a newswire archive from millions of scans of local newspapers across a century. The dataset spans from 1878 - the very early days of newswires - through 1977 - when a copyright law change resulted in the content no longer being in the public domain. `Newswire` contains digitized texts of 2.7 million unique, de-duplicated newswire articles. While we have no way to know if this is every article that ran over the wire - since not every local paper that ever existed and subscribed to the wire has been preserved - it covers a vast diversity of news.

Our deep learning pipeline first detects layouts and transcribes over 138 million front page articles from U.S. local newspapers, spanning all 50 states. It then uses a contrastively trained syntactic similarity model (22) to accurately determine which articles come from the same underlying newswire source article, in the presence of significant abridgement and noise. Articles are georeferenced, and topics are tagged using customized neural topic classifiers. Moreover, named entities are tagged and classified into different entity types, and people are disambiguated to Wikipedia and Wikidata using a custom neural disambiguation model. This open-source pipeline provides a blueprint for how an archive can be reconstructed from dispersed noisy reproductions of texts using deep learning, a relevant task for a variety of applications in the digital humanities and computational social sciences.

Each article appears once in `Newswire`, although some were reproduced over a hundred times in our corpus. De-duplicating the dataset is important for its utility for language model training, as generative language models are exponentially more likely to regenerate content that is duplicated in their training corpus (15; 14; 13). It also makes the dataset much smaller and easier to work with for researchers in the social sciences, humanities, and computational linguistics. For researchers interested in which papers ran a given article on their front page, we provide their Library of Congress metadata. This provides information on the location of each newspaper, dates of publication, and merges/splits of newspapers over time. These will allow researchers to examine the social, political, and economic factors that drove papers to choose certain content for reproduction from the overall newswire menu that they could access.

While we would like to provide a dataset that extends through the present, copyright law changes prevent this. Newswire articles are in the public domain because, until the latter part of the 20th century, texts had to be published with a copyright notice and copyrights renewed to remain under copyright, which was a costly process. (16) documents that newswire articles are not under copyright in the period we consider, as yesterday's news had no economic value to justify the costs of copyrighting it. This is why the dataset ends in 1977, when a change to copyright law made this content automatically copyrighted. Some other types of reproduced content, such as serialized fiction, could still be under copyright if written later in the period. To ensure that non-wire content (which is quite distinct linguistically) is removed, we run a highly accurate text classifier that determines whether a reproduced front-page article comes from a newswire or another syndicated source. The vast majority of reproduced front-page content, especially later in the period, comes from newswires.

In addition to the digitized texts, the structured information from georeferencing (18,209 distinct locations), topic classification (34 topics), named entity recognition (43,759,476 named entity mentions),

and entity disambiguation (61,933 unique disambiguated individuals) can facilitate applications ranging from knowledge intensive natural language processing to historical scholarship.

The dataset is on Hugging Face[1], with a CC-BY license. This will facilitate language modeling applications, ranging from tuning existing models to training a purely historical language model for specialized applications. In addition to providing data for language model training and historical language modeling, we hope it will make it easier for researchers in the social sciences and humanities to work with historical newspaper data at scale. To this end, we will provide tutorial notebooks.

The rest of this paper is organized as follows. Section 2 reviews the related literature, and Section 3 describes `Newswire`. Section 4 outlines and evaluates the methods used to construct `Newswire`, and Section 6 considers limitations.

## 2   Related Literature

Constructing `Newswire` required customized, cheaply scalable methods for layout recognition, OCR, content association, georeferencing, topic classification, named entity recognition, and entity disambiguation, drawing on a variety of literatures.

Digitization builds on literatures on object detection (12; 25) and OCR (6; 5). Digitized newspaper collections exist for many countries, but do not provide information on reproduced content. There is nevertheless a literature on detecting reproduced texts. The Viral Texts project (23) was designed for detecting reproduced content in the Library of Congress's Chronicling America collection, a primary source of image scans for `Newswire`. Library of Congress provides a page-level OCR (that does not detect individual headlines, articles, captions, etc. and sometimes scrambles content). The Viral Texts pipeline looks for overlapping $n$-gram spans, with much of its complexity tailored towards Library of Congress's messy page level OCR.

In contrast, (22) detect reproduced content by contrastively training a neural bi-encoder model on hand-labeled, paired articles from newswires. The encoder maps articles from the same newswire article source to similar embeddings and articles from different sources to different embeddings. These embeddings are then grouped with highly scalable single linkage clustering to identify reproduced content. (9) show that the neural detection of reproduced text - combined with structured article rather than page level data - outperforms the viral texts n-gram approach by a wide margin. Hence, we use our model from (22) to detect reproduced articles for `Newswire`.

For topic tagging, we draw upon the open-source comparative agendas project (4), whose New York Times Index dataset contains short synopses and corresponding topic labels for a sampling of articles that ran in the New York Times between 1946 and 2016. We build on this work, extending the comparative agendas classification system to `Newswire`. Because their dataset includes only very short synopses of the articles, which are out-of-distribution from the full texts in `Newswire`, we obtain full texts and merge them with the labeled synopses using a Sentence-BERT MPNet semantic similarity model (17). The labeled full texts are then used to train the multiclass topic classifier.

Finally, we expand upon the literature on entity disambiguation to disambiguate individuals in `Newswire` to Wikipedia. In particular, our contrastively trained bi-encoder architecture builds upon (26), incorporating various advances from the past five years, novel training data drawn from Wikipedia disambiguation pages, and entities that are not in the knowledgebase, a common feature in the real world `Newswire` data.

The most closely related dataset to `Newswire` is Headlines (21), which provides paired headlines from post-1920 wire articles. Headlines for wire articles were written by local papers, and hence different headlines were used to describe the same wire article. Headlines is designed for semantic similarity applications. It does not include the article texts, or any of the structured information we extract from these texts. `Newswire` also relates to the American Stories dataset (9), which includes over 438 million digitized article texts from Library of Congress's Chronicling America collection of newspaper scans. Reproduced articles are not extracted and it likewise does not include georeferenced datelines, topic tags, tagged entities, or disambiguated entities.

---

[1] https://huggingface.co/datasets/dell-research-harvard/newswire

# 3   Dataset

The texts in `Newswire` are constructed by detecting noisily reproduced articles from nearly 138M digitized front page newspaper articles, spanning 1878 to 1977. There are 99M unique articles, with around 2.9M reproduced more than three times. We use this threshold because in practice, articles reproduced less typically resulted from duplicate scans of the same newspaper. 2.7M of these articles are newswires, reproduced around 32M times. Table 1 provides summary statistics.

| Description | Count |
|---|---|
| Front page articles | 137,941,190 |
| Unique articles (including singletons) | 99,472,910 |
| Unique articles reproduced >3 times | 2,889,012 |
| Unique wire articles | 2,719,607 |
| Total reproductions of wire articles | 32,107,676 |

Table 1: Counts of articles meeting various criteria in our raw digitized newspaper corpus.

Figure 1 shows the distribution of reproduced wire articles across time. The peak in the 1950s represents the zenith of print news, before its market share was eroded by television news. The shift in 1920 is due to the smaller size of our corpus prior to this date. We have continued to expand our underlying American Stories corpus (9) since publication, as Library of Congress adds more scans to its collection, and we will update `Newswire` accordingly as well.

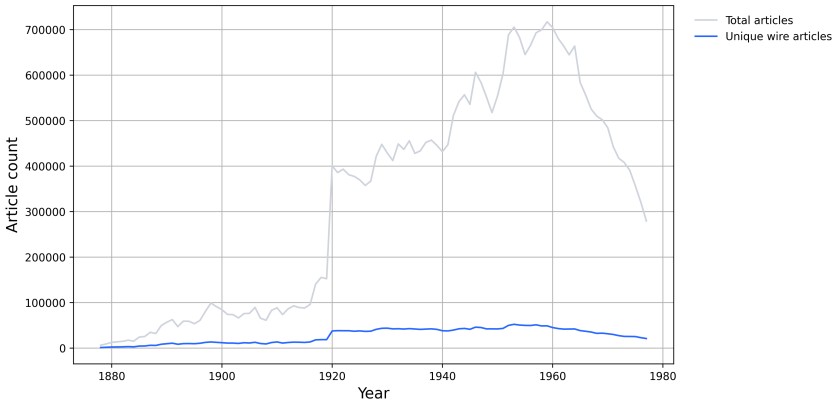

Figure 1: Counts of total articles and unique newswire articles in `Newswire`.

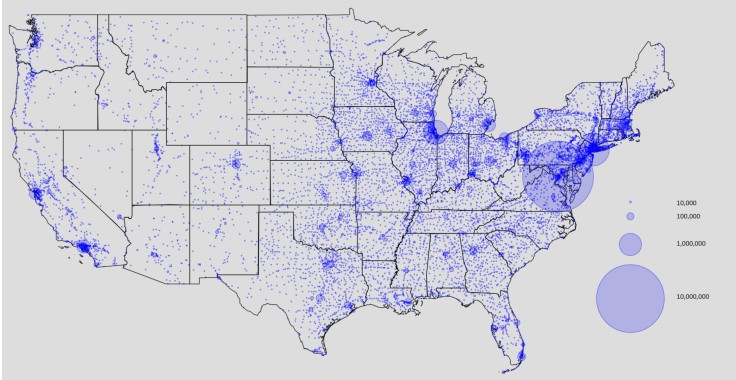

Figure 2: Reproduction of newswire articles with domestic datelines.

For each unique newswire article, we use sophisticated pipelines to impute a variety of structured information. First, the articles are georeferenced. The dataset contains datelines from 18,209 unique georeferenced locations, the most common of which is Washington DC (27% of content), followed

by New York (5%) and London (5%). 25.7% of articles have international datelines over the period, peaking during the World Wars. Figure 2 shows a map of the georeferenced content in the U.S., with the area of the dot proportional to the number of times that an article with that dateline is reproduced. While Washington stands out, so does the broad coverage. Figure 3 plots the share of international datelines, as a fraction of all datelines, across time. Europe in general receives much more coverage than other regions, with exceptions during the Korean and Vietnam Wars.

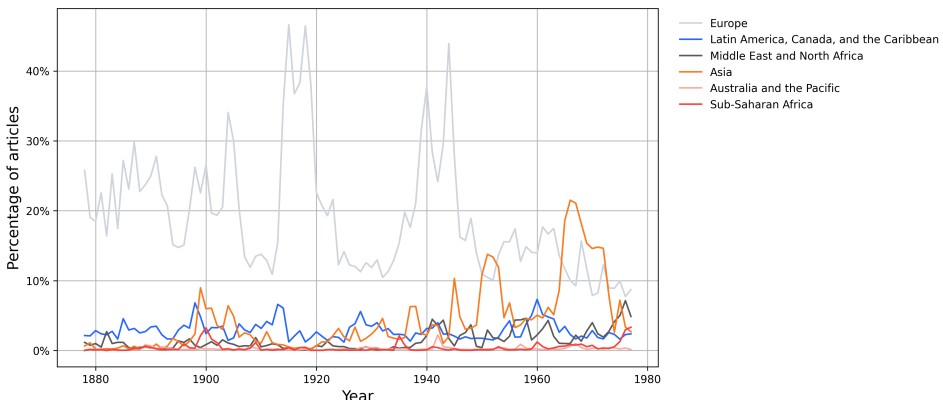

Figure 3: Reproduction of newswire articles with international datelines.

`Newswire` also includes tags for seven topics, proportions of which are shown in figure 4. These topics were chosen as they are of general interest across the period. The most common topic is politics, encompassing around 37% of articles. The data pass basic sanity checks. For example, crime coverage is elevated during Prohibition, and coverage of protests and the Civil Rights Movement peak in the 1960s.

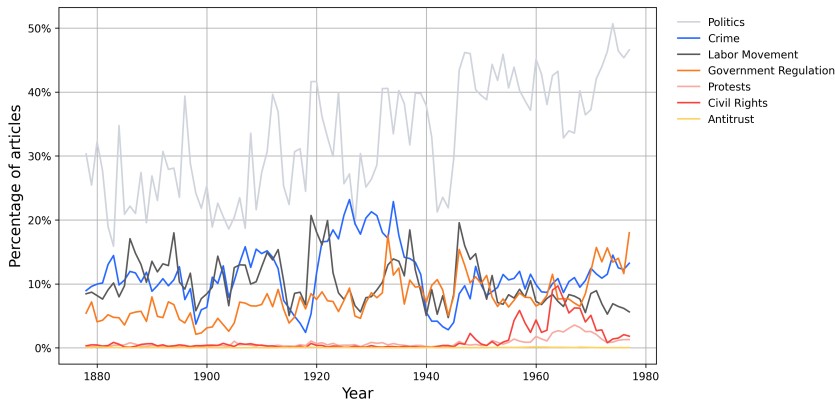

Figure 4: Share of reproduced newswire articles with a given binary topic tag, across time.

We also include a set of multiclass topic tags, which tag which of the policy agendas from the Comparative Agendas project most accurately describes the topic of the article (4). The distribution of these topic tags is shown in Figure 5.

We also tag 43.7 million entity mentions: people, locations, organizations, and other miscellaneous proper nouns. We select these entity types following standard practice in the NER literature. These are, for example, the same entity types that are labeled in CoNLL-2003 (18). These entities are reproduced nearly 596 million times in our underlying article corpus. Figure 6 plots the share of entities in different categories (person, location, organization, miscellaneous) across time. World War II is again evident, with the spike of locations and miscellaneous entities (e.g., named aircraft).

We disambiguate 15,323,463 person mentions - comprising 61,933 unique individuals - to Wikidata. The most mentioned entity is Dwight D. Eisenhower, who is mentioned in 9,530 unique articles reproduced an average of 33.7 times. Richard Nixon, Harry S. Truman, Adolf Hitler and Nikita

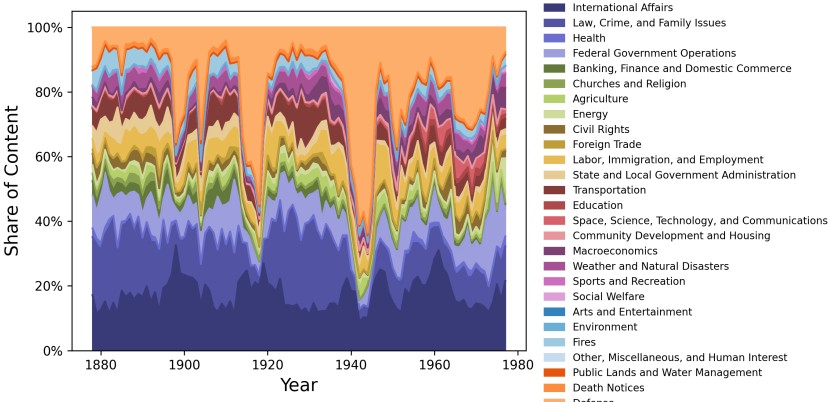

Figure 5: The distribution of multiclass topic tags, trained on data from the Comparative Agendas project.

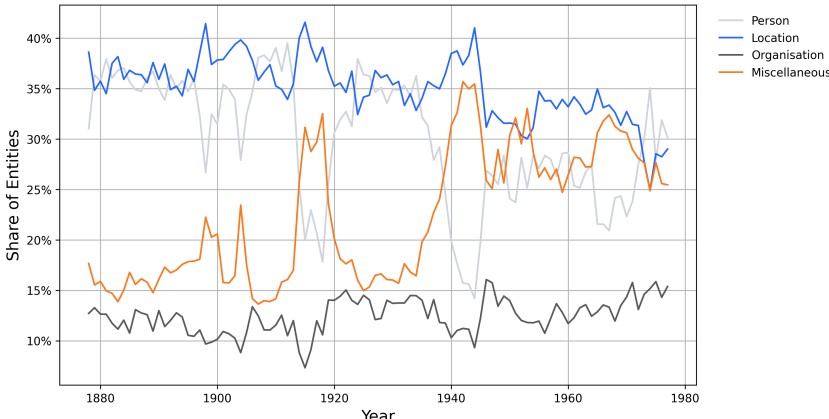

Figure 6: Share of entities by type: person, location, organization, and miscellaneous.

Khrushchev are also amongst the top-5 most mentioned. Only 4.6% of disambiguated entity mentions refer to women, and the most mentioned woman is Golda Meir. Among the most common occupations are politician, military officer and lawyer. There is a rich variety of additional structured data on these individuals available via the Wikidata tags which researchers using `Newswire` can explore.

## 4  Methods and Evaluation

**Digitization:** Creating a comprehensive newswire archive from dispersed reproduced content required digitizing structured article texts from newspaper image scans. We recognize newspaper layouts (*e.g.,* headlines, articles, captions, images, ads, headers, etc.) in off-copyright local newspapers, using object detection (25; 12), trained on labeled images. Newspaper layouts are extremely heterogeneous. To obtain training data from the tails of the layout object distribution, we developed an active learning method (20) that selects objects to label using a perturbation-based uncertainty scoring method. The layout analysis achieved a mean average precision of 93.31 on articles and is documented and evaluated in more detail in (9).

Before transcribing the article texts, we classify which ones are legible using an image classifier. We also developed a novel OCR architecture (6) and OCR package (5) - tuned to the newspaper texts - to obtain a computationally efficient, high quality OCR. (Post-1920 content was digitized prior to this with Tesseract.) These steps are also described in more detail in the appendix. Articles with texts spanning multiple bounding boxes need to be associated, for which we use a customized RoBERTa

cross-encoder model to predict whether the content in one bounding box continues the content in another, as described in (21).

**De-duplication of content:** This pipeline results in 138M digitized front page article texts, spanning the period from 1878 to 1977, and we next need to detect which of these are noisy reproductions. This is a non-trivial task, since articles were edited and often significantly abridged - both by the regional newswire bureaus and by local newspapers - and can contain OCR and layout detection noise. We created a dataset of nearly 123,000 hand-labeled positive wire article pairs that come from the same newswire source. Hard negatives were defined as articles with a high *n-gram* similarity that did not come from the same underlying source, *e.g.,* articles about the same event from different newswires and updates about unfolding news stories. We contrastively tuned a Sentence-BERT (17) MPNet (24) bi-encoder model, using an online contrastive loss, to map articles from the same article source to similar representations and articles from different sources to dissimilar representations. We then apply single linkage clustering, which is highly scalable. This achieves an adjusted rand index of 91.5, in contrast to 73.7 for locality-sensitive hashing (the leading scalable sparse method for detecting reproduced texts) (22). Our chosen method can be slightly improved upon by adding a cross-encoder on top of the bi-encoder, as shown in table 3, but as the gains from doing this are minimal, we choose not to include this step, for computational efficiency. All technical details are provided in the appendix.

**Detection of wire content:** This method detects any content that is reproduced, including articles which did not come from a newswire. The non-newswire reproduced content primarily consists of templates used by our off-copyright local papers to report various recurrent local news: *e.g.,* the roster for the high school football game, the schedule of church services, the weather forecast, etc. We use a distil-RoBERTa classifier to accurately remove weather forecasts. Local news templates, in contrast, are vast in their diversity, complicating their removal via a neural classifier. Fortunately, this pattern of reproduction makes them simple to remove in post-processing. News is timely, and newswires were reproduced within a narrow time window (typically hours) of each other. In contrast, local templates such as sports rosters tend to contain a diversity of dates in the cluster of reproduced articles, and some of the reproductions are coming from the same newspaper, which is again not characteristic of newswires. Hence, to remove this content, we use simple rules based on the diversity of dates and whether the same paper is reproducing a given piece of content (see (21) for details). Finally, there are other types of nationally syndicated content, such as opinion and lifestyle columns and serialized fiction. These have very different linguistic styles from news and rarely appear on the front page. We remove them with a distil-RoBERTa classifier that achieves nearly 96% accuracy. Full details of this are given in the appendix. Of the articles that remain after post-processing, nearly 95% are wire articles, with this percentage even higher later in the period.

Text quality can vary across reproduced articles; we choose the version of the article with the modal number of paragraphs with the lowest non-word rate (in the SymSpell dictionary[2]) for inclusion in `Newswire`. In experiments, we found that large language models such as ChatGPT and Claude did an excellent job in many cases of removing remaining OCR errors. While we do not have the resources to run these models over the 2.7 million unique articles in `Newswire`, we are working on tuning an open-source model to achieve similar performance and plan to include the cleaned texts in the next `Newswire` release.

**Georeferencing:** A key component of `Newswire` is the various structured information that we impute. Datelines - which give the location where the article is written - are georeferenced to GeoNames coordinates. Rather than relying on a single article from each reproduced cluster, we individually detect potential datelines for each article in a cluster before aggregating these predictions together. We begin by finetuning a DistilBERT model to reliably extract bylines from the text of each article. For each article in the cluster, we then match all potential $n$-grams within the article's byline to the set of cities, states, and countries present in the GeoNames dataset that have at least 500 residents. Finally, we aggregate these article-level predictions to generate a final dateline for each cluster of reproduced articles. On a test set of 2,324 hand-labeled georeferenced tags, we find that the pipeline has an accuracy of 94.9%. Further details of this method and the test set are given in the appendix. XX Comparison to other method XX

**Topic tagging:** We compute two different sets of topic tags using fine-tuned neural classifiers. First, we hand-labeled training data for 7 topics, shown in the top of table 4, and train customized classifiers.

---

[2]`https://github.com/wolfgarbe/SymSpell`

Articles were double-labeled, with discrepancies resolved by hand. We began with these particular topics because of their centrality to a variety of social science questions. Performance is evaluated in table 2. We found that RoBERTa large classifiers generally achieved the best performance and ran those over the 2.7 million articles in `Newswire`. Full training details are in the appendix.

| Topic | Train size | Eval size | Test size | F1 |
|-------|-----------|-----------|-----------|-----|
| Politics | 2418 | 498 | 1473 | 84.9 |
| Crime | 463 | 98 | 98 | 90.4 |
| Labor movement | 253 | 54 | 54 | 94.1 |
| Government regulation | 612 | 131 | 131 | 87.5 |
| Protests | 351 | 75 | 75 | 90.6 |
| Civil rights | 943 | 202 | 202 | 87.0 |
| Antitrust | 329 | 70 | 70 | 93.8 |
| Sports | 339 | 72 | 72 | 94.1 |
| Fires | 554 | 118 | 118 | 97.1 |
| Weather and natural disasters | 574 | 122 | 122 | 92.3 |
| Death notices | 272 | 57 | 57 | 100 |

Table 2: Topic classifier performance

Second, we train a multiclass classifier to associate each article with a policy topic from the Comparative Agendas project (4). Training data for this classifier comes from the Comparative Agenda project (full details are in the appendix). The sports, fires, weather and death notices categories have a small number of labels and perform poorly, so we replace them with our customized binary classifiers, which are described in the bottom of table 2. The results of this process were evaluated on randomly selected hand-labeled `Newswire` articles, with an accuracy rate of 87%.

**Named entity recognition:** We tag all entities in the articles using named entity recognition (NER). Our model, trained on double-labeled entity data from historical newspapers, achieves an F1 of 90.4 in correctly identifying spans of text containing named entities without regards to the class, outperforming a Roberta-Large model fine-tuned on CoNLL03 by a large margin. It identifies people with an F1 of 94.3 (10). The supplementary materials provide additional data and training details. Table 3 compares this custom classifier to an off-the-shelf NER model, which achieves an F1 of only 77.8 on the test set.

**Entity disambiguation:** Finally, we disambiguate entities to Wikipedia/Wikidata. Off-the-shelf models did not perform well on this task (as shown in table 3), as many can disambiguate only to the most common Wikidata entities (e.g., (27)), do not handle entities that are not in the knowledgebase (the main benchmark, AIDA-CONLL, only has entities in the knowledgebase so this is common), or are extremely computationally intensive to run (e.g., (27)). Instead, we train customized entity co-reference and disambiguation bi-encoder models on Wikipedia disambiguation pages.

First, we train a model to resolve co-references across wire articles within dates. We start with a base S-BERT MPNet bi-encoder model (17). This is constrastively trained on 179 million pairs taken from mentions of entities on Wikipedia, where positives are mentions of the same individual. Hard negatives are mined using individuals that appear on the same disambiguation pages. Embeddings from the tuned co-reference resolution model are then clustered using Hierarchical Agglomerative Clustering. We average across embeddings of co-referenced entities, to form a prototype.

We then use a fine-tuned entity disambiguation model to disambiguate these prototypes to Wikipedia first paragraphs of people. This is trained again on Wikipedia data, in this case mentions paired with Wikipedia first paragraphs. It is then further tuned on a novel hand-labelled newspaper dataset, described in more detail in the supplementary materials.

For inference, we merge Wikipedia to Wikidata records that have instance type `<human>` and contain either a birth or death date, limiting the database to people. This results in 1,118,257 unique records as our knowledge base. We then use qrank to assign more popular entities to the final result in case of very close matches.

To evaluate the model, we label 157 wire articles, from 4 different days from 4 years, totalling 1,137 person mentions. We chose days on which State of the Union addresses took place, as there are more coreferences to resolve on these days, providing more power for evaluating this task. For the

coreference step, we achieve an ARI of 98.2. Overall, we associate the correct entity in 72.9% of cases. In cases where we make an association, it is correct in 96.8% of cases, and in cases where there was no corresponding entity in the knowledge base, we correctly do not make an association in 95.4% of cases. Therefore the majority of the errors are cases where there was an entity in the knowledge base, but we predict no association. Extra details on training, datasets and sensitivity to hyperparameters are given in the appendix.

| Pipeline Stage | Model | Statistic | Result | Used |
|---|---|---|---|---|
| OCR | EffOCR-Word (Small) (5) | CER | 0.015 | * |
| | Tesseract OCR | CER | 0.106 | * |
| | EasyOCR | CER | 0.170 | |
| | PaddleOCR | CER | 0.304 | |
| Deduplication | Locally Sensitive Hashing | ARI | 73.7 | |
| | N-gram Overlap | ARI | 75.0 | |
| | Neural biencoder (22) | ARI | 91.5 | * |
| | Biencoder + crossencoder | ARI | 93.7 | |
| Georeferencing | N-gram matching | Accuracy | 94.9 | * |
| | GPT-4o-mini | Accuracy | 85.3 | |
| NER | Custom NER (10) | F1 | 90.4 | * |
| | Roberta-Large tuned on CoNLL03 (7) | F1 | 77.8 | |
| Entity disambiguation | LinkNewsWikipedia (1) | Accuracy | 78.3 | * |
| | BLINK (26) | Accuracy | 59.9 | |
| | GENRE (8) | Accuracy | 63.4 | |
| | ReFinED (2) | Accuracy | 65.4 | |

Table 3: Models considered for each stage of the pipeline and comparisons of their performance on the test set for each stage. Starred models are those used. For CER, smaller values are better, while for the other statistics, bigger numbers are better. CER = Character Error Rate, ARI = Adjusted Rand Index.

## 5  Applications

Newswire has a diversity of uses, ranging from language model training to social science, computational linguistics, and digital humanities scholarship. At the same time, newspapers can be analogized as a first (albeit incomplete) draft of history. There are a variety of settings where including Newswire in an LLM training corpus would be useful. Not everything in that first draft ends up preserved in an online database such as Wikipedia today, and so seeing this data in training would expose an LLM to additional information. The information could also be useful as part of an external database in a retrieval augmented language modeling setup.

There are other motivations for exposing an LLM to historical training data as well. For instance, (19) trains a language model from scratch on the American Stories historical newspaper dataset (9) to avoid "look-ahead" bias when evaluating whether an LLM can make financial predictions about the future with past data. Training on historical data can also reduce copyright risk in an uncertain legal environment.

Newswire is also relevant for studying linguistic change across time, and for a diversity of questions in social science and the digital humanities. Many social scientists and historians use historical newspapers for granular information about the past. (3) provides a recent review. Most extant datasets can only be access via keyword search; indeed, (3) dedicate much attention to how to conduct keyword searches effectively. By providing full text data, Newswire will allow social scientists and historians to carry out far more rich analyses of the contents of historical news. By posting the dataset on hugging face and providing tutorials, we aim to lower the costs of using Newswire in scholarly research.

# 6 Limitations and Recommended Usage

`Newswire` is ethically sound. It is limited to news that was of regional or national importance historically, and hence does not contain private information. While there are many potential uses of `Newswire`, however, there are also features that may make it unsuitable for certain applications. Historical texts reflect the linguistic and cultural norms of the times and places where they were written. For many scholarly applications, this is what make the dataset useful and interesting. It also broadens the diversity of cultural contexts that a language model could be exposed to. At the same time, some of the content may be inaccurate or considered offensive. We have not filtered it for toxicity, as this would invalidate the use of the data for social science research questions. Moreover, while the OCR is generally reasonable, the dataset is also not suited for contexts requiring completely clean texts. Rather, `Newswire` has a range of applications from language model training to social science research to serving as a repository of knowledge that is of interest to the general public. The open-source pipelines used to digitize, tag, and disambiguate the data also provide an accessible blueprint for curating large-scale historical text datasets.

## Acknowledgements:

Microsoft Azure, the Harvard Data Science Initiative, and the Harvard Economics Department Ken Griffin Fund provided resources to digitize the local newspaper content. Yiyang Chen and Katherine Liu provided excellent research assistance. We would also like to acknowledge our collaborators on the development of methods that made constructing this dataset possible, who are cited in the main text.

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
