# Supplementary Materials

# 1   Code Availability

All code used to create `Newswire` is available at our public Github repository.

# 2   Model Details

## 2.1   Digitalization

To identify content regions like articles, headlines, and ads in newspaper scans, we employed YOLOv8 (Medium) (20), starting from the official YOLOv8m pretrained model. We trained for over 100 epochs on 2,202 scans with 48,874 layout objects. This model achieves a 0.91 mAP50:95 for articles and 0.84 mAP50:95 for headlines. The confidence threshold was lowered to 0.1 to enhance recall.

Text bounding boxes were classified as legible, borderline, or illegible using MobileNetV3 (Small) (10), initialized from a PyTorch Image Models checkpoint (22). Training involved 979 examples (678 legible, 192 borderline, 109 illegible) over 50 epochs with weighted Cross Entropy Loss and a learning rate of 2e-3, which was reduced every 20 epochs, as described in (6).

Text prior to 1920 was OCRd using EffOCR (5; 3), while later text was transcribed using Tesseract.

Articles with texts spanning multiple bounding boxes need to be associated, for which we use a customized RoBERTa cross-encoder model to predict whether the content in one bounding box continues the content in another, as described in (17).

## 2.2   De-duplication of content

To detect reproduced content, we use a bi-encoder model, as developed by (18). This model, based on the S-BERT MPNET model (16; 19), is fine-tuned to learn similar representations for reproduced articles and dissimilar representations for non-reproduced ones. The model is fine-tuned on a manually labeled dataset of near-duplicate articles. The model is fine-tuned with a learning rate of 2e-05, using the online contrastive loss implementation from S-BERT (9), for 16 epochs, with a batch size of 32 and a 100% warm-up. A cosine similarity metric with a 0.2 margin is used. The model achieves an ARI of 91.5 on a hand-labeled test set of over 100 million pairs of articles (54,996 positives pairs and 100,914,159 negative pairs). This performance is compared to other baselines in table **??** in the main text.

Once the embeddings are generated, to create clusters of near duplicates we apply highly scalable single-linkage clustering, setting a cosine similarity threshold of 0.94. Articles are represented as nodes within a graph, and edges are formed when the cosine similarity surpasses the threshold. Edge weights are calculated based on the negative exponential of the time gap (in days) between the articles. To ensure clusters are meaningful, we use Leiden community detection, which helps mitigate the risk of false-positive edges merging unrelated articles into the same cluster.

To further refine the clustering, we exclude any clusters containing more than 50 articles that span over five different dates. Similarly, clusters with more than 50 articles are removed if the number of articles exceeds twice the count of unique newspapers from which they originate. These criteria ensure the exclusion of clusters that, although correctly grouped based on a shared source, are not useful for the `Newswire` dataset.

A detailed analysis of errors is provided in (18). Typically errors are articles about the same story from different wire sources, or updates to a story as new events unfolded.

## 2.3   Detection of wire content

To accurately filter out non-wire content, we fine-tuned a Distil-RoBERTa classifier on a hand-labeled training set of 1,459 samples. The model was trained for 20 epochs with a batch size of 64 and a

learning rate of $5\mathrm{e}{-}5$ with an AdamW optimizer. All hyperparameters were selected based on the model's performance on a validation set containing 336 labeled samples. The final model achieved an F1 of 0.96 on a test set containing 448 samples.

## 2.4 Georeferencing

Our georeferencing pipeline consists of multiple steps designed to extract the dateline from each cluster of reproduced articles. As a first step, we train a DistilBERT classifier to detect bylines from each article on a training set of 1,392 hand-labeled samples. The model was trained for 25 epochs with a batch size of 16 and a learning rate of $2\mathrm{e}{-}5$ with an AdamW optimizer. All hyperparameters were selected based on the model's performance on a validation set containing 464 labeled samples. The final byline classifier achieved an F1 of 0.92 on a test set containing 464 samples.

For each article within a given cluster, we take all possible $n$-grams from the detected bylines, matching each consecutive sequence of words to GeoNames' dictionary of city and country names. We additionally detect state names and state abbreviations within bylines. We first search for matches among capitalized $n$-grams, as most datelines in our corpus are capitalized, searching across all $n$-grams only in the event that we do not find a match.

Once we have potential matches for each article in a cluster, we aggregate these matches to get a tentative match for the city, state (if one exists), and country in each cluster dateline. For both state and country, we take the most common potential match across all articles in the cluster. As some city names may be substrings of other city names (for example, York and New York), we additionally weight the count of each potential city match by a function of the length of the city name. In all cases, if the tentative match fails to appear in at least $15\%$ of all articles in the cluster, we proceed without a tentative match; this is to prevent the pipeline from detecting errant place names in clusters with no dateline. The AP stylebook additionally designates a list of 56 cities which are allowed to appear in AP articles without an associated state/country name – to address these cases, we manually match these cities to their associated states/countries.

Having a tentative match for the city, state, and country in which each article cluster was written, we attempt to merge these tentative matches with GeoNames' dataset of all cities with a population of at least 500 residents. Some datelines that contain locations other than cities, such as the Johnson Space Center, or very sparsely populated areas may fail to be matched as a result of this process. After running the georeferencing pipeline over our entire sample, we manually inspected the matches for any particularly common instances of these non-city datelines. We include further explanation of these exceptions in the "wire_location_notes" field associated with the cluster.

On a test set of 2,324 hand-labeled georeferenced clusters, we find that the pipeline has an accuracy of 94.9%.

### 2.4.1 Benchmarking against GPT-4o-mini

We additionally benchmark our georeferencing pipeline against GPT-4o-mini, passing in the following prompt:

I will feed you the beginning snippet of multiple articles belonging to a given cluster – in a cluster, articles should all be the same. If there is a geographic byline belonging to the articles in a cluster, I would like you to output the location. If it is in the United States, please give me the city name, state, and country. If it is not in the United States, please give me both the city name and the country name.

Some articles in the cluster may have a byline while others may not – if there are multiple different locations, please output only the one you believe is correct. Only output locations that correspond to the article byline – if there are other articles mentioned in the text but that are not part of a byline, ignore these. Please output only a single location and nothing else. If there is no location, output None.

For example, the following snippet: "In Vienna, Austria, there is much indignation beeause in the Balkan states a monument has been erected in honor of the student" has no location in the byline – Vienna does not belong to a byline. You should output None.

Meanwhile, the snippet: "LOS ANGELES, Jan. 27.–The appeal of Alexander Pantages to his conviction on charges of having assaulted" has Los Angeles in the byline. You should output Los Angeles, California, United States.

Remember, please output only a single location and nothing else. If there is no location, output None.

The above prompt achieves an accuracy of 85.3% on the same test set of 2,324 hand-labeled georeferenced clusters, compared to our pipeline's accuracy of 94.9%.

## 2.5 Topic tagging

Two types of topics are tagged in the dataset.

First, we tag topics of particular interest during this period (Politics, Crime, Labor movement, Government regulation, Protests, Civil rights, Antitrust). To create training data for these models, we developed a pipeline to efficiently extract articles, as random sampling would not lead to many on topic articles. We did this in two steps. First, we trained a BART-large (13) bi-encoder on MNLI (23), using the Dense Passage Retrieval (DPR) infrastructure (12). We trained for 40 epochs, with a batch size of 32, and a learning rate of 7e-05. This is a re-ranking model, so at inference time, it ranks all embedded texts with respect to a query text. We embedded all `Newswire` articles with this model, and formatted queries as "this example is about topic" (e.g., "this example is about civil rights"). From the results, we extracted the highest scoring articles. We run zero-shot classification (using Huggingface's implementation, based on bart-large-mnli (13)) to classify whether these texts were on topic or not, compared to the same query. We then sampled from the on-topic and not on-topic predictions to create our datasets. These datasets were then manually labeled. For each topic we then trained a binary topic classifier. Table 1 gives hyperparameters for each model. The size of the labeled datasets and the evaluation results on the test set are shown table in the main text.

The second type of classifier is a multi-class classifier, which categorises data into the classes from the Comparative Agendas project (2) (30 major policy topics, such as Labor, Immigration, and Employment, Education, Environment, Energy, Immigration, Transportation). To train this, we use data from the Comparative Agenda project, as they have already labeled 4,026 short article synopses from the New York Times according to these policy topics. As we wanted to train on articles, not on synopses, we use a semantic similarity model (S-BERT MPNet) to match these synopses to the articles that they are summarising. We are able to match 1847 articles, and this match has a top-1 retrieval accuracy of 95%, evaluated over 44 articles. These 1847 articles form our training data for this multi-class classifier. We used these to fine-tuned a RoBERTA-large model, for 4 epochs with a batch size of 32 and a learning rate of 5e-5. We found that the results of this classifier for four topics (sports, fires, weather and natural disasters, and death notices) were poor, due to a small amount of labeled data. So in these four cases, we replaced the labels with the results of binary classifiers trained on these topics, using the same process as for the other binary classifiers. The results of this second classification process were evaluated on randomly selected hand-labeled `Newswire` articles, with an accuracy rate of 87%.

## 2.6 Named Entity Recognition

Off-the-shelf NER did not perform satisfactorily on this data, so we trained a custom model. For training data, we randomly selected articles from `Newswire`, which were hand-labelled. These data are described in table 2. All data were double-labeled by two highly-trained undergraduate research assistants, and all discrepancies were resolved by hand. Annotator instructions are reproduced in full in (7). We used these to fine-tune a Roberta-Large model (14) for 184 epochs, with a batch size

| Topic | Base model | Learning rate | Batch size | Epochs |
|---|---|---|---|---|
| Politics | RoBERTa-large | 1e-6 | 8 | 50 |
| Crime | RoBERTa-large | 1e-6 | 8 | 50 |
| Labor movement | distilRoBERTa-base | 1e-5 | 32 | 50 |
| Government regulation | RoBERTa-large | 5e-6 | 8 | 50 |
| Protests | distilRoBERTa-base | 1e-5 | 32 | 50 |
| Civil rights | RoBERTa-large | 1e-5 | 8 | 50 |
| Antitrust | RoBERTa-large | 1e-5 | 8 | 50 |
| Sports | RoBERTa-large | 1e-6 | 8 | 50 |
| Fires | distilRoBERTa-base | 5e-6 | 16 | 30 |
| Weather and natural disasters | distilRoBERTa-base | 5e-6 | 16 | 30 |
| Death notices | RoBERTa-large | 1e-5 | 8 | 50 |

Table 1: Topic classifier training details

of 128, and a learning rate of 4.7e-05. Table 2 describes the training data and performance. These results are benchmarked in table **??** in the main text.

| Entity Type | Data | | | Evaluation | | |
|---|---|---|---|---|---|---|
| | Train | Eval | Test | Precision | Recall | F1 |
| Location | 1191 | 192 | 199 | 87.4 | 94.5 | 90.8 |
| Misc | 1037 | 149 | 181 | 73.7 | 68.6 | 79.6 |
| Organisation | 450 | 59 | 83 | 80.7 | 80.7 | 80.7 |
| Person | 1345 | 231 | 261 | 92.9 | 95.8 | 94.3 |

Table 2: NER data and performance

## 2.7 Entity Disambiguation

To disambiguate entities to Wikidata/Wikipedia we start with the NER output and subset it to [PER] (person) tags since we are most interested in them. We then collect each named entity within and across all newspaper articles on a given day and run it through our customized entity coreference pipeline to collapse all entity mentions on a given day into a single prototype (cluster of mentions). We use this prototype to disambiguate the constituent mentions to the entity's Wikidata ID.

We imagine entity coreference and disambiguation as semantic textual similarity tasks. Entity coreference can be seen as linking similar entity mentions, and disambiguation as linking an entity mention to a template created by Wikipedia and Wikidata. The template is constructed using the entity's name, alias, and occupation from Wikidata and concatenating it with the entity's first paragraph in Wikipedia. Semantic similarity is measured by information that is encoded by custom contrastively trained bi-encoder models based on Sentence Transformers (16).

We process a Wikipedia XML dump [1] from November 11, 2022, and collect mentions of each entity (that appears as a hyperlink in the dump). We then split entities into a train-test-val split and pair up mentions of the same entity and associated context (defined by the paragraph containing the entity mention). These are positive pairs. We pair up an entity mention with mentions of another entity to form 'easy negatives'. We augment our training data by adding 'hard' negatives where we use a novel approach of using disambiguation pages from Wikipedia that contain confusables of popular entities in the Wikiverse. For instance, the disambiguation page "John Kennedy" contains, John F. Kennedy the president, John Kennedy (Louisiana politician) (born 1951), a United States Senator from Louisiana, and John F. Kennedy Jr. (1960–1999), son of President Kennedy. We sample some contexts where John F. Kennedy was mentioned and pair them up with a context around a mention of an entity within a disambiguation page and treat this as a hard negative pair. We found that the performance of our models improved a lot by having a decorator or a set of special tokens ($[M]$ Entity $[\backslash M]$ around an entity mention (24). For example, consider this context about President Kennedy

---
[1] https://dumps.wikimedia.org/

"Eisenhower sharing a light moment with President-elect $[M]$ John F. Kennedy $[\backslash M]$ during their meeting in the Oval Office at White House". Some contexts naturally have multiple entities, like "Eisenhower" and "John F. Kennedy" in this case. We found that we can improve the features of these special tokens by further augmenting our training data with in-context negatives - pairing up these contexts with multiple negatives that only differ in the placement of the special tokens. With all of the variants ready we have, 179069981, 5819525, and 5132565 train, val, and test pairs respectively. We use a sequence length of 256 and truncate contexts around the mentions when necessary. We start with an *all-mpnet-base-v2* model sourced from the Hugging Face hub (21) and fine-tune it using these pairs. We train the model in Pytorch (15) with hyperparameters tuned with hyperband implemented within Weights and Biases (1).

We use Online Contrastive Loss as implemented in (16) and use AdamW as the optimizer with a linear warmup scheduler (20%). We train on 4 Nvidia A6000 GPUs with a batch size of 512, a learning rate of 1e-5, and a contrastive margin of 0.4. We run it for only a single epoch - seeing each pair in the train split only once. The best model is selected using pair-wise classification F1 on the validation set (the best val F1 was 92.75%). With a large dataset like this, we found it useful to divide it into 10 chunks before we began training. After finishing each chunk (1/10 of an epoch), since we resumed training on an intermediate checkpoint, we lowered the learning rate to 2e-6 after the first chunk, to reduce the chances of the optimizer overshooting the minima. Because training each chunk started with a warmup, effectively, our strategy simulated a linear scheduler with restarts.

Once the model is trained we embed all the newspaper articles and cluster the embeddings of articles printed on the same date using Hierarchical Agglomerative Clustering implemented with Scikit-Learn (4) with average linkage, cosine metric, and a threshold of 0.15. The clusters from this exercise are essentially mentions of the same entity on a given day. We average the embeddings within a cluster to create entity prototypes for each date. We will use these prototypes for disambiguation.

Next, we prepare a lookup corpus for disambiguating entity mentions (or prototypes) to the right entity using semantic information from both the context around the mention and information about it from a template we create. To create the template, we obtained names, aliases, and occupations/positions held by individuals from Wikidata. Consider the example of President Kennedy - "'John F. Kennedy is of type human. Also known as Kennedy, Jack Kennedy, President Kennedy, John Fitzgerald Kennedy, J. F. Kennedy, JFK, John Kennedy, John Fitzgerald "Jack" Kennedy, and JF Kennedy. Has worked as politician, journalist, statesperson". We then suffix this template with the first paragraph of the associated Wikipedia page.

Next, we adapt our coreference model for the disambiguation task. We link up the contexts with entity mentions with the associated entity template to form positive pairs. Easy negatives link contexts with random entity templates. As with our coreference training, we utilize Wikipedia disambiguation pages and family information from wiki data to associate entity contexts with hard negative templates. We then split entities in an 80-10-10 train-val-test split ending up with 4202145, 522385, and 528709 pairs in the respective split. We fine-tune our coreference model with similar hyperparameters as the coreference training, except without restarts (or chunking) and with the learning rate of 2e-6, batch size of 256, and 20% warmup. The model was trained for 1 epoch and the best checkpoint was selected using classification F1 as before (max validation F1 was 97%). Since the disambiguation of newspapers to the knowledge base is our main task, we adapt the training domain further to newspapers. We prepare a gold dataset to fine-tune the model on pairs crafted from newspaper contexts and Wikipedia templates. First, we obtained the names and aliases of individuals from Wikidata. Then, we search for them in our newspaper corpus, hand labeling whether they refer to the person searched for. When they do not match, these form hard negatives. We form extra hard negatives by matching an entity with another entity mentioned in the same context. We also form Wikipedia hard negatives by matching an entity with another entity mentioned in the same Wikipedia disambiguation dictionary. Finally, we create easy negatives by matching with a random entity. This dataset is described in table 3. We start with the model trained on Wikipedia pairs and fine-tune the model with an identical training setup. The maximum validation F1 achieved was 85%.

| Split | Positives | Easy negatives | Hard Negatives | Wikipedia hard negatives |
|---|---|---|---|---|
| Train | 1426 | 1299 | 1460 | 861 |
| Eval | 189 | 175 | 184 | 118 |
| Test | 198 | 180 | 183 | 130 |

Table 3: Data for finetuning entity disambiguation

At inference time, we prune our knowledge base to remove extraneous entities. First, we only keep those entities that have either a birth or a death date. Second, we only keep those people born before 1970 (considering the period of our data). If the birth date was missing, the entity was retained. Finally, we remove those entities having no overlap and a high edit distance between the Wikidata label and the associated Wikipedia page's title - this allows us to keep only those Wikidata entities whose Wikipedia page corresponds to the actual entity and not something related to it. Our pruning exercise brings the total number of entities in our knowledge base from 1.8 million to about 1.12 million. We then embed the templates of these entities using our fine-tuned disambiguation model and stored them in an FAISS IndexFlatIP index (11). Since our embeddings are normalized, Inner Product boils down to Cosine Similarity. We then use the date-entity clusters obtained before and embed the mentions within each cluster using the model trained for disambiguation, average them (within-cluster), create entity-date prototype embeddings, and treat them as queries. To improve the quality of our results, we utilize Qrank [2] which ranks Wikidata entities by aggregating page views on Wikipedia, Wikispecies, Wikibooks, Wikiquote, and other Wikimedia projects. We first retrieve the 10 nearest neighbors of each query. We keep only those neighbors that are at most 0.01 Cosine Distance away from the nearest match. We then use Qrank to rerank these results, essentially preferring the popular entity in cases where the returned matches are very close to each other. The Wikidata ID of the nearest embedding (after re-ranking) is then assigned to the date-entity cluster associated with the query, essentially disambiguating the clusters as well as their constituents to Wikidata. This of course is akin to treating disambiguation as a semantic retrieval problem and not handling out-of-knowledge-base entities. Our architecture allows us to use the Cosine Similarity between the entity-date prototype and the nearest template to evaluate whether or not the entity is an acceptable match. Anything lower than the threshold can be considered as either an incorrect match or out of the knowledge base. We tune a no-match threshold using a sample of human-annotated data from the `Newswire`. We annotate the output of our disambiguation pipeline on a set of 6,425 pairs sampled from 13 years - as correct if the returned entity is correct and incorrect when it is not. We then find the cut-off threshold that maximizes pair-wise classification precision and use that as the no-match threshold.

---

[2] `https://github.com/brawer/wikidata-qrank/tree/main`

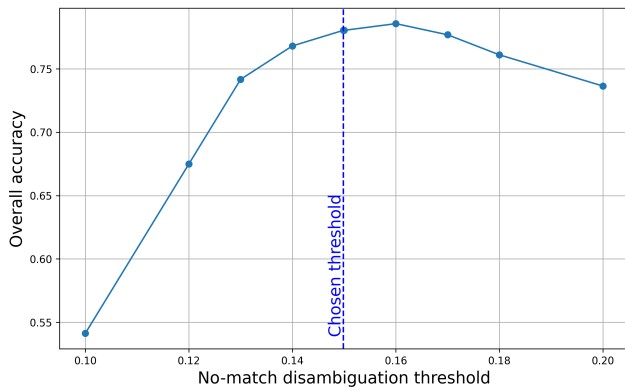

Figure 1: Sensitivity of disambiguation results to choice of no-match threshold.

## 2.8 Models and Dataset

We have made our models (see Table 4) and training/evaluation data available on the Hugging Face hub for reproducibility and ease of access by other practitioners.

| Repo Name | Content |
| --- | --- |
| dell-research-harvard/NewsWire | The `Newswire` dataset |
| dell-research-harvard/historical_newspaper_ner | NER model for Historical Newspapers |
| dell-research-harvard/LinkMentions | Coreference model trained on Wikipedia |
| dell-research-harvard/LinkWikipedia | Disambiguation model trained on Wikipedia |
| dell-research-harvard/NewsLinkWikipedia | Disambiguation model fine-tuned on newspapers |
| dell-research-harvard/topic-politics | Topic model for politcs |
| dell-research-harvard/topic-crime | Topic model for crime |
| dell-research-harvard/topic-labor-movement | Topic model for the labor movement |
| dell-research-harvard/topic-govt-regulation | Topic model for government regulation |
| dell-research-harvard/topic-protests | Topic model for protests |
| dell-research-harvard/topic-civil-rights | Topic model for civil rights |
| dell-research-harvard/topic-antitrust | Topic model for antitrust |
| dell-research-harvard/topic-sports | Topic model for sports |
| dell-research-harvard/topic-fires | Topic model for fires |
| dell-research-harvard/topic-weather | Topic model for weather and natural disasters |
| dell-research-harvard/topic-obits | Topic model for death notices |
| dell-research-harvard/byline-detection | Byline detection model |
| dell-research-harvard/wire-classifier | Classifier for wire articles |

Table 4: Models and Dataset on the Hugging Face Hub

# 3 Dataset Information

## 3.1 Dataset URL

`Newswire` can be found at `https://huggingface.co/datasets/dell-research-harvard/newswire`.

## 3.2 DOI

The DOI for this dataset is: 10.57967/hf/2423.

## 3.3 Metadata URL

Croissant metadata for `Newswire` can be found at `https://huggingface.co/api/datasets/dell-research-harvard/newswire/croissant`.

## 3.4 License

`Newswire` has a Creative Commons CC-BY license.

## 3.5 Dataset usage

The dataset is hosted on huggingface, in json format. Each year in the dataset is divided into a distinct file (eg. 1952_data_clean.json).

An example from `Newswire` looks like:

```
{
    "year": 1880,
    "dates": ["Feb-23-1880"],
```

```
"article": "SENATE Washington, Feb. 23.--Bayard moved that in respect of the
    memory of George Washington the senate adjourn ... ",
"byline": "",
"newspaper_metadata": [
    {
        "lccn": "sn92053943",
        "newspaper_title": "the rock island argus",
        "newspaper_city": "rock island",
        "newspaper_state": " illinois "
    },
    ...
],
"antitrust": 0,
"civil_rights": 0,
"crime": 0,
"govt_regulation": 1,
"labor_movement": 0,
"politics": 1,
"protests": 0,
"ca_topic": "Federal Government Operations",
"ner_words": ["SENATE", "Washington", "Feb", "23", "Bayard", "moved", "that",
    "in", "respect", "of", "the", "memory", "of", "George", "Washington",
    "the", "senate", "adjourn", ... ],
"ner_labels": ["B-ORG", "B-LOC", "O", "B-PER", "B-PER", "O", "O", "O", "O",
    "O", "O", "O", "O", "B-PER", "I-PER", "O", "B-ORG", "O", ...],
"wire_city": "Washington",
"wire_state": "district of columbia",
"wire_country": "United States",
"wire_coordinates": [38.89511, -77.03637],
"wire_location_notes": "",
"people_mentioned": [
    {
        "wikidata_id": "Q23",
        "person_name": "George Washington",
        "person_gender": "man",
        "person_occupation": "politician"
    },
    ...
],
"cluster_size": 8
}
```

The data fields are:

- year: year of article publication.

- dates: list of dates on which this article was published, as strings in the form mmm-DD-YYYY.

- byline: article byline, if any.

- article: article text.

- newspaper_metadata: list of newspapers that carried the article. Each newspaper is repre-
sented as a list of dictionaries, where lccn is the newspaper's Library of Congress identifier,
newspaper_title is the name of the newspaper, and newspaper_city and newspaper_state
give the location of the newspaper.

314  - `antitrust`: binary variable. 1 if the article was classified as being about antitrust.

315  - `civil_rights`: binary variable. 1 if the article was classified as being about civil rights.

316  - `crime`: binary variable. 1 if the article was classified as being about crime.

317  - `govt_regulation`: binary variable. 1 if the article was classified as being about government
318  regulation.

319  - `labor_movement`: binary variable. 1 if the article was classified as being about the labor movement.

320  - `politics`: binary variable. 1 if the article was classified as being about politics.

321  - `protests`: binary variable. 1 if the article was classified as being about protests.

322  - `ca_topic`: predicted Comparative Agendas topic of article.

323  - `wire_city`: City of wire service bureau that wrote the article.

324  - `wire_state`: State of wire service bureau that wrote the article.

325  - `wire_country`: Country of wire service bureau that wrote the article.

326  - `wire_coordinates`: Coordinates of city of wire service bureau that wrote the article.

327  - `wire_location_notes`: Contains wire dispatch location if it is not a geographic location. Can
328  be one of "Pacific Ocean (WWII)", "Supreme Headquarters Allied Expeditionary Force (WWII)",
329  "North Africa", "War Front (WWI)", "War Front (WWII)" or "Johnson Space Center".

330  - `people_mentioned`: list of disambiguated people mentioned in the article. Each disambiguated
331  person is represented as a dictionary, where `wikidata_id` is their ID in Wikidata, `person_name` is
332  their name on Wikipedia, `person_gender` is their gender from Wikidata and `person_occupation`
333  is the first listed occupation on Wikidata.

334  - `cluster_size`: Number of newspapers that ran the wire article. Equals length of
335  `newspaper_metadata`.

336  The whole dataset can be easily downloaded using the `datasets` library:

```
337  from datasets import load_dataset
338  dataset_dict = load_dataset("dell-research-harvard/newswire")
```

339  Specific files can be downloaded by specifying them:

```
340  from datasets import load_dataset
341  load_dataset(
342      "dell-research-harvard/newswire",
343      data_files=["1929_data_clean.json", "1969_data_clean.json"]
344  )
```

### 3.6 Author statement

346  We bear all responsibility in case of violation of rights.

### 3.7 Hosting, licensing and maintenance Plan

348  We have chosen to host `Newswire` on huggingface as this ensures long-term access and preservation
349  of the dataset.

### 3.8 Dataset documentation and intended uses

351  We follow the datasheets for datasets template (8).

### 3.8.1 Motivation

**For what purpose was the dataset created?** Was there a specific task in mind? Was there a specific gap that needed to be filled? Please provide a description.

*The dataset was created to provide researchers with a large, high-quality corpus of structured and transcribed newspaper article texts from American newswires. These texts provide a massive repository of information about historical topics and events. The dataset will be useful to a wide variety of researchers including historians, other social scientists, and NLP practitioners.*

**Who created this dataset (e.g., which team, research group) and on behalf of which entity (e.g., company, institution, organization)?**
*`Newswire` was created by a team of researchers at Harvard University.*

**Who funded the creation of the dataset?** If there is an associated grant, please provide the name of the grantor and the grant name and number.

*The creation of the dataset was funded by the Harvard Data Science Initiative, and the Harvard Economics Department Ken Griffin Fund. Compute credits provided by Microsoft Azure to the Harvard Data Science Initiative.*

**Any other comments?**
*None.*

### 3.8.2 Composition

**What do the instances that comprise the dataset represent (e.g., documents, photos, people, countries)?** Are there multiple types of instances (e.g., movies, users, and ratings; people and interactions between them; nodes and edges)? Please provide a description.

*`Newswire` comprises instances of newspaper articles. Accompanying each article is a list of newspapers that ran the article, classification of whether the article is about certain topics, a list of entities detected in the article, and a disambiguation of people mentioned in the article.*

**How many instances are there in total (of each type, if appropriate)?**
*`Newswire` contains 2,719,607 unique articles.*

**Does the dataset contain all possible instances or is it a sample (not necessarily random) of instances from a larger set?** If the dataset is a sample, then what is the larger set? Is the sample representative of the larger set (e.g., geographic coverage)? If so, please describe how this representativeness was validated/verified. If it is not representative of the larger set, please describe why not (e.g., to cover a more diverse range of instances, because instances were withheld or unavailable).

*Many newspapers were not preserved, so we cannot guarantee that this dataset contains all possible instances.*

**What data does each instance consist of? "Raw" data (e.g., unprocessed text or images) or features?** In either case, please provide a description.

*Each data instance consists of raw data and dervied data. Specifically, an example from `Newswire` is:*

```
{
    "year": 1880,
    "dates": ["Feb-23-1880"],
    "article": "SENATE Washington, Feb. 23.--Bayard moved that in respect of the
```

```
394            memory of George Washington the senate adjourn ... ",
395        "byline": "",
396        "newspaper_metadata": [
397            {
398                "lccn": "sn92053943",
399                "newspaper_title": "the rock island argus",
400                "newspaper_city": "rock island",
401                "newspaper_state": " illinois "
402            },
403            ...
404        ],
405        "antitrust": 0,
406        "civil_rights": 0,
407        "crime": 0,
408        "govt_regulation": 1,
409        "labor_movement": 0,
410        "politics": 1,
411        "protests": 0,
412        "ca_topic": "Federal Government Operations",
413        "ner_words": ["SENATE", "Washington", "Feb", "23", "Bayard", "moved", "that",
414            "in", "respect", "of", "the", "memory", "of", "George", "Washington",
415            "the", "senate", "adjourn", ... ],
416        "ner_labels": ["B-ORG", "B-LOC", "O", "B-PER", "B-PER", "O", "O", "O", "O",
417            "O", "O", "O", "O", "B-PER", "I-PER", "O", "B-ORG", "O", ...],
418        "wire_city": "Washington",
419        "wire_state": "district of columbia",
420        "wire_country": "United States",
421        "wire_coordinates": [38.89511, -77.03637],
422        "wire_location_notes": "",
423        "people_mentioned": [
424            {
425                "wikidata_id": "Q23",
426                "person_name": "George Washington",
427                "person_gender": "man",
428                "person_occupation": "politician"
429            },
430            ...
431        ],
432        "cluster_size": 8
433    }
```

The data fields are:

- `year`: year of article publication.

- `dates`: list of dates on which this article was published, as strings in the form mmm-DD-YYYY.

- `byline`: article byline, if any.

- `article`: article text.

- `newspaper_metadata`: list of newspapers that carried the article. Each newspaper is represented as a list of dictionaries, where `lccn` is the newspaper's Library of Congress identifier, `newspaper_title` is the name of the newspaper, and `newspaper_city` and `newspaper_state` give the location of the newspaper.

- `antitrust`: binary variable. 1 if the article was classified as being about antitrust.

- `civil_rights`: binary variable. 1 if the article was classified as being about civil rights.

- `crime`: binary variable. 1 if the article was classified as being about crime.

- `govt_regulation`: binary variable. 1 if the article was classified as being about government regulation.

- `labor_movement`: binary variable. 1 if the article was classified as being about the labor movement.

- `politics`: binary variable. 1 if the article was classified as being about politics.

- `protests`: binary variable. 1 if the article was classified as being about protests.

- `ca_topic`: predicted Comparative Agendas topic of article.

- `wire_city`: City of wire service bureau that wrote the article.

- `wire_state`: State of wire service bureau that wrote the article.

- `wire_country`: Country of wire service bureau that wrote the article.

- `wire_coordinates`: Coordinates of city of wire service bureau that wrote the article.

- `wire_location_notes`: Contains wire dispatch location if it is not a geographic location. Can be one of "Pacific Ocean (WWII)", "Supreme Headquarters Allied Expeditionary Force (WWII)", "North Africa", "War Front (WWI)", "War Front (WWII)" or "Johnson Space Center".

- `people_mentioned`: list of disambiguated people mentioned in the article. Each disambiguated person is represented as a dictionary, where `wikidata_id` is their ID in Wikidata, `person_name` is their name on Wikipedia, `person_gender` is their gender from Wikidata and `person_occupation` is the first listed occupation on Wikidata.

- `cluster_size`: Number of newspapers that ran the wire article. Equals length of `newspaper_metadata`.

**Is there a label or target associated with each instance?**    If so, please provide a description.

*The data is not labelled, but has had inference from multiple models run on it.*

**Is any information missing from individual instances?**    If so, please provide a description, explaining why this information is missing (e.g., because it was unavailable). This does not include intentionally removed information, but might include, e.g., redacted text.

*In some cases, there may be no* `byline`*, as the article did not have one, or it was not detected.* `wire_city`*,* `wire_state`*,* `wire_country`*,* `wire_coordinates` *are missing when no location was detected.* `person_gender` *and* `person_occupation` *are missing if no gender or occupation was listed on Wikidata.*

**Are relationships between individual instances made explicit (e.g., users' movie ratings, social network links)?**    If so, please describe how these relationships are made explicit.

*No relationships between instances are made explicit.*

**Are there recommended data splits (e.g., training, development/validation, testing)?**    If so, please provide a description of these splits, explaining the rationale behind them.

*There are no recommended splits.*

**Are there any errors, sources of noise, or redundancies in the dataset?**    If so, please provide a description.

*The data is sourced from OCR'd text of historical newspapers. Therefore some of the texts contain OCR errors.*

**Is the dataset self-contained, or does it link to or otherwise rely on external resources (e.g., websites, tweets, other datasets)?** If it links to or relies on external resources, a) are there guarantees that they will exist, and remain constant, over time; b) are there official archival versions of the complete dataset (i.e., including the external resources as they existed at the time the dataset was created); c) are there any restrictions (e.g., licenses, fees) associated with any of the external resources that might apply to a future user? Please provide descriptions of all external resources and any restrictions associated with them, as well as links or other access points, as appropriate.

*The data is self-contained.*

**Does the dataset contain data that might be considered confidential (e.g., data that is protected by legal privilege or by doctor-patient confidentiality, data that includes the content of individuals non-public communications)?** If so, please provide a description.

*The dataset does not contain information that might be viewed as confidential.*

**Does the dataset contain data that, if viewed directly, might be offensive, insulting, threatening, or might otherwise cause anxiety?** If so, please describe why.

*The headlines in the dataset reflect diverse attitudes and values from the period in which they were written, 1878-1977, and contain content that may be considered offensive for a variety of reasons.*

**Does the dataset relate to people?** If not, you may skip the remaining questions in this section.

*Many news articles are about people.*

**Does the dataset identify any subpopulations (e.g., by age, gender)?** If so, please describe how these subpopulations are identified and provide a description of their respective distributions within the dataset.

*The dataset does not specifically identify any subpopulations.*

**Is it possible to identify individuals (i.e., one or more natural persons), either directly or indirectly (i.e., in combination with other data) from the dataset?** If so, please describe how.

*If an individual appeared in the news during this period, then article text may contain their name, age, and information about their actions. Further, for prominent individuals, we have disambiguated them to Wikipedia, which directly identifies them.*

**Does the dataset contain data that might be considered sensitive in any way (e.g., data that reveals racial or ethnic origins, sexual orientations, religious beliefs, political opinions or union memberships, or locations; financial or health data; biometric or genetic data; forms of government identification, such as social security numbers; criminal history)?** If so, please provide a description.

*All information that it contains is already publicly available in the newspapers used to create the data.*

**Any other comments?**
*None.*

### 3.8.3 Collection Process

**How was the data associated with each instance acquired?** Was the data directly observable (e.g., raw text, movie ratings), reported by subjects (e.g., survey responses), or indirectly inferred/derived from other data (e.g., part-of-speech tags, model-based guesses for age or language)? If data was reported by subjects or indirectly inferred/derived from other data, was the data validated/verified? If so, please describe how.

*The dataset combines raw data and derived data. The pipeline used to extract the data and to create the derived data is described in detail within the paper. The dataset described here is the output of that pipeline.*

**What mechanisms or procedures were used to collect the data (e.g., hardware apparatus or sensor, manual human curation, software program, software API)?** How were these mechanisms or procedures validated?

*These methods are described in detail in the main text and supplementary materials of this paper.*

**If the dataset is a sample from a larger set, what was the sampling strategy (e.g., deterministic, probabilistic with specific sampling probabilities)?**
*The dataset was not sampled from a larger set.*

**Who was involved in the data collection process (e.g., students, crowdworkers, contractors) and how were they compensated (e.g., how much were crowdworkers paid)?**
*We used student annotators to create the validation and test sets. They were paid $15 per hour, a rate set by a Harvard economics department program providing research assistantships for undergraduates.*

**Over what timeframe was the data collected? Does this timeframe match the creation timeframe of the data associated with the instances (e.g., recent crawl of old news articles)?** If not, please describe the timeframe in which the data associated with the instances was created.

*The articles were written between 1878 and 1977. They were processed between 2020 and 2024.*

**Were any ethical review processes conducted (e.g., by an institutional review board)?** If so, please provide a description of these review processes, including the outcomes, as well as a link or other access point to any supporting documentation.

*No, this dataset uses entirely public information and hence does not fall under the domain of Harvard's institutional review board.*

**Does the dataset relate to people?** If not, you may skip the remaining questions in this section.

*Historical newspapers contain a variety of information about people.*

**Did you collect the data from the individuals in question directly, or obtain it via third parties or other sources (e.g., websites)?**
*The data were obtained from historical newspapers.*

**Were the individuals in question notified about the data collection?** If so, please describe (or show with screenshots or other information) how notice was provided, and provide a link or other access point to, or otherwise reproduce, the exact language of the notification itself.

*Individuals were not notified; the data came from publicly available newspapers.*

**Did the individuals in question consent to the collection and use of their data?** If so, please describe (or show with screenshots or other information) how consent was requested and provided, and provide a link or other access point to, or otherwise reproduce, the exact language to which the individuals consented.

*The dataset was created from publicly available historical newspapers.*

**If consent was obtained, were the consenting individuals provided with a mechanism to revoke their consent in the future or for certain uses?** If so, please provide a description, as well as a link or other access point to the mechanism (if appropriate).

*Not applicable.*

**Has an analysis of the potential impact of the dataset and its use on data subjects (e.g., a data protection impact analysis) been conducted?** If so, please provide a description of this analysis, including the outcomes, as well as a link or other access point to any supporting documentation.

*No.*

**Any other comments?**
*None.*

### 3.8.4 Preprocessing/cleaning/labeling

**Was any preprocessing/cleaning/labeling of the data done (e.g., discretization or bucketing, tokenization, part-of-speech tagging, SIFT feature extraction, removal of instances, processing of missing values)?** If so, please provide a description. If not, you may skip the remainder of the questions in this section.

*See the description in the main text.*

**Was the "raw" data saved in addition to the preprocessed/cleaned/labeled data (e.g., to support unanticipated future uses)?** If so, please provide a link or other access point to the "raw" data.

*All data is in the dataset.*

**Is the software used to preprocess/clean/label the instances available?** If so, please provide a link or other access point.

*No specific software was used to clean the instances.*

**Any other comments?**
*None.*

### 3.8.5 Uses

**Has the dataset been used for any tasks already?** If so, please provide a description.

*No.*

**Is there a repository that links to any or all papers or systems that use the dataset?** If so, please provide a link or other access point.

*No such repository currently exists.*

**What (other) tasks could the dataset be used for?**
*There are a large number of potential uses in the social sciences, digital humanities, and deep learning research, discussed in more detail in the main text.*

**Is there anything about the composition of the dataset or the way it was collected and preprocessed/cleaned/labeled that might impact future uses?** For example, is there anything that a future user might need to know to avoid uses that could result in unfair treatment of individuals or groups (e.g., stereotyping, quality of service issues) or other undesirable harms (e.g., financial harms, legal risks) If so, please provide a description. Is there anything a future user could do to mitigate these undesirable harms?

*This dataset contains unfiltered content composed by newspaper editors, columnists, and other sources. It reflects their biases and any factual errors that they made.*

**Are there tasks for which the dataset should not be used?** If so, please provide a description.

*We would urge caution in using the data to train generative language models - without additional filtering - as it contains content that many would consider toxic.*

**Any other comments?**
*None*

### 3.8.6 Distribution

**Will the dataset be distributed to third parties outside of the entity (e.g., company, institution, organization) on behalf of which the dataset was created?** If so, please provide a description.

*Yes. The dataset is available for public use.*

**How will the dataset will be distributed (e.g., tarball on website, API, GitHub)** Does the dataset have a digital object identifier (DOI)?

*The dataset is hosted on huggingface. Its DOI is 10.57967/hf/2423.*

**When will the dataset be distributed?**
*The dataset was distributed on 7th June 2024.*

**Will the dataset be distributed under a copyright or other intellectual property (IP) license, and/or under applicable terms of use (ToU)?** If so, please describe this license and/or ToU, and provide a link or other access point to, or otherwise reproduce, any relevant licensing terms or ToU, as well as any fees associated with these restrictions.

*The dataset is distributed under a Creative Commons CC-BY license. The terms of this license can be viewed at* `https: // creativecommons. org/ licenses/ by/ 2. 0/`

**Have any third parties imposed IP-based or other restrictions on the data associated with the instances?** If so, please describe these restrictions, and provide a link or other access point to, or otherwise reproduce, any relevant licensing terms, as well as any fees associated with these restrictions.

*There are no third party IP-based or other restrictions on the data.*

**Do any export controls or other regulatory restrictions apply to the dataset or to individual instances?** If so, please describe these restrictions, and provide a link or other access point to, or otherwise reproduce, any supporting documentation.

*No export controls or other regulatory restrictions apply to the dataset or to individual instances.*

**Any other comments?**
*None.*

### 3.8.7 Maintenance

**Who will be supporting/hosting/maintaining the dataset?**

*The dataset is hosted on huggingface.*

**How can the owner/curator/manager of the dataset be contacted (e.g., email address)?**

*The recommended method of contact is using the huggingface 'community' capacity. Additionally, Melissa Dell can be contacted at melissadell@fas.harvard.edu.*

**Is there an erratum?** If so, please provide a link or other access point.

*There is no erratum.*

**Will the dataset be updated (e.g., to correct labeling errors, add new instances, delete instances)?** If so, please describe how often, by whom, and how updates will be communicated to users (e.g., mailing list, GitHub)?

*If we update the dataset, we will notify users via the huggingface Dataset Card.*

**If the dataset relates to people, are there applicable limits on the retention of the data associated with the instances (e.g., were individuals in question told that their data would be retained for a fixed period of time and then deleted)?** If so, please describe these limits and explain how they will be enforced.

*There are no applicable limits on the retention of data.*

**Will older versions of the dataset continue to be supported/hosted/maintained?** If so, please describe how. If not, please describe how its obsolescence will be communicated to users.

*If we update the dataset, older versions of the dataset will not continue to be hosted. We will notify users via the huggingface Dataset Card.*

**If others want to extend/augment/build on/contribute to the dataset, is there a mechanism for them to do so?** If so, please provide a description. Will these contributions be validated/verified? If so, please describe how. If not, why not? Is there a process for communicating/distributing these contributions to other users? If so, please provide a description.

*Others can contribute to the dataset using the huggingface 'community' capacity. This allows for anyone to ask questions, make comments and submit pull requests. We will validate these pull requests. A record of public contributions will be maintained on huggingface, allowing communication to other users.*

**Any other comments?**
*None.*