# OpenReview forum: "Newswire: A Large-Scale Structured Database of a Century of Historical News"
_NeurIPS.cc/2024/Datasets_and_Benchmarks_Track — NeurIPS 2024 Track Datasets and Benchmarks Poster_

### Official Review · Reviewer_GpZ3 · 2024-07-17
**Interesting database of US wire service news stories**

**Rating:** 7
**Confidence:** 4
**Correctness:** The approaches and results appear cor…
**Clarity:** The paper is well organized and motiv…

**Review:**

This paper introduces an interesting database of US wire-service news
stories, It is effective in showing the AI techniques used to generate the
database and that the database is broadly useful.  Please see the individual
sections for additional comments, concerns, and opportunities for improvment.

**Strengths:**

Key strengths come from the authors' combined mastery of AI techniques,
archival information, and the likely needs of users of this archival
information.  The result is a high-quality database useful to a broad variety
of academic and public policy areas.  The authors appear to have considered
all of the key factors necessary for high-quality digitization and devised
effective techniques to deal with them.  And I thought the original idea
clever:  use newspaper front pages to create this database of wire stories.

The authors also took great care to ensure that their source material was
legally available to place in the public domain via the Newswire database.
That effort included both limiting the years covered and in the techniques
deployed to ensure that no non-public domain material crept in.

I also applaud the authors' decision not to censor any of the material, so
that users of Newswire can achieve the most accurate assessments of
historical information.

**Additional Feedback:**

No additional comments.

**Documentation:**

Appropriate documentation and background is given.

**Ethics:**

The authors have taken considerable care in addressing ethics and law.  I see no issues.

**Limitations:**

As noted above in the "Opportunities for Improvement", it would be nice
to see some comment on additional implications and limitations of the work.
However, in the main, the authors seem to have covered key limitations.

**Opportunities For Improvement:**

Most of the opportunities for improvement are places where it would be
interesting and helpful in terms of limitations of the database to know more
than the paper provided -- perhaps due to space constraints.

For example, it would be useful to note what wire services actually appeared
and in what fractions.  For example, it seems likely that virtually all
stories from 1877 - 1907 were from Associated Press (since United Press and
the International Service) did not come into being until then.  If it is the
case that virtually all wire service stories in the early period were from
AP, did the multiplicity of wire services available after 1907 impact the
quality or breadth of digitization?

Given the efficacy of the techniques used, how easy would it be to apply them
to other countries?  In addition to potential legal issues and obtaining
front-page scans, there is also the amount of human manual effort to do
appropriate tests and calibration.  How much manual effort was required in
total to produce Newswire for the United States?

Another question is why were only newspaper front pages used in finding wire
service stories, and not the rest of the paper?  Is there an estimate (e.g.
from sampling a small subset) of what fraction of wire service stories
appeared on front pages (of at least some newspapers).

Similarly, it would be useful to comment if any key stories or important
aspects of life were missed by the focus on wire service stories.  Beyond any
front-page bias in Newswire, through most of their history prior to the
Internet age, newspapers relied heavily on revenue from advertising:  large
firms and local merchants offering their wares plus classified ad placements
from individuals.  These ads consumed a considerable fraction of the physical
page space.  To this day, there are disputes about whether "public notice"
statements must be printed in newspapers, or whether other approaches can be
used, e.g. is placing these notices on a website sufficient and more
appropriate in the modern era?  US state legislatures wrestle with this
decision -- with a large part of the back-and-forth based on the revenue loss
to newspapers were such "public notices" to go elsewhere.  Is it possible to
quantify what fraction of newspaper information is captured in the Newswire
database?  Could the Newswire techniques also be used to capture some of
these other non-wire-service domains?

The results of investigations based on Newswire might even impact some
legislative decisions on matters like "public notice".  How will future
historians learn about and study our period?  Do "public notices" and other
material provide a key archival resource for the future that may be lost if
notices are places are placed on a more evanescent or ephemeral medium like a
website.

**Relation To Prior Work:**

Prior work is covered well.

**Summary And Contributions:**

The paper introduces Newswire, a carefully crafted database of wire service
stories appearing in US newspapers from 1878-1977 -- where the primary wire
services are Associated Press (established 1846), United Press (established
1907) and the International Service (established 1909 and merged with United
Press in 1958).  Other non-US-based wire services like Reuters or Agence
France Presse may also present.  The wire services themselves have digitized
archives, but the paper observes that the digitization is of low quality, and
the results are not easily accessible by the broad public.

To do better, the authors started with a repository of scans of more than 138
million US newspaper front pages during this period.  The paper details a
variety of AI techniques used to perform key digitization tasks, e.g. OCR,
named entity resolution, recognition of duplicates, etc.  Much of this effort
is automated, but the paper notes a number of places where manual techniques
were used to calibrate, assess, and guide the automation.  Error rates are
provided for each task, and range from mid eighties to high nineties.  The
result is a well-digitized searchable database suitable for use by
historians, sociologists, public policy officials, and others.  The paper
provides evidence of the database's utility with a variety of examples
outlining broad trends in news coverage over time.  These trends and charts
show the value of this well-crafted database.  Because the end result is a
database with tags for each news story, the broad power of SQL can be used to
extract all manner of information and correlations -- something not as easily
possible with a mere set of digitized articles.

---

> ### Author Rebuttal · Authors · 2024-08-16
>
> Many thanks for your thoughtful comments. We are glad that you found the paper interesting. Below, respond to your individual points and suggestions.
>
> *It would be useful to note what wire services actually appeared and in what fractions. For example, it seems likely that virtually all stories from 1877 - 1907 were from Associated Press (since United Press and the International Service) did not come into being until then. If it is the case that virtually all wire service stories in the early period were from AP, did the multiplicity of wire services available after 1907 impact the quality or breadth of digitization?*
>
> We are in the process of detecting which newswire articles in Newswire are from. This requires re-running customized OCR, as ‘AP’, ‘UPI’ etc. are written in special ligatures. We will add this information to a subsequent update of Newswire. While the number of articles per year does increase after 1907, we have seen no evidence to suggest that there is any impact on the quality of digitalisation.
>
> *Given the efficacy of the techniques used, how easy would it be to apply them to other countries? In addition to potential legal issues and obtaining front-page scans, there is also the amount of human manual effort to do appropriate tests and calibration. How much manual effort was required in total to produce Newswire for the United States?*
>
> While we spent a significant amount of time and effort producing Newswire, our publicly available pipeline and datasets should make it much easier to produce a similar resource for another country. For example, because of transfer learning, our layout analysis model could likely be tuned to other contexts with much more limited training data than we needed to train it using an ImageNet pre-trained backbone. Furthermore, our datasets could be machine translated if necessary to provide training data for tasks like de-duplication and entity disambiguation in other languages. (Of course, there are also a variety of other countries that have English as an official language.) In short, Newswire provides a valuable resource for anyone looking to extend historical news archives in the future.
>
>
> *Why were only newspaper front pages used in finding wire service stories, and not the rest of the paper? Is there an estimate (e.g. from sampling a small subset) of what fraction of wire service stories appeared on front pages (of at least some newspapers).*
>
> We chose to focus on front pages due to resource constraints. Even with extremely lightweight layout detection and OCR, the cost to digitise the full editions for all the papers in our database would have been around half a million dollars. Our digitisation was funded by regularly acquiring small amounts of academic credits from Microsoft Azure over the past five years and with some very limited resources from our department. We continue to seek a larger grant to digitise full editions. With that said, given the thousands of newspapers in our corpus, we think that it is very likely that the vast majority of stories that went out over the national wire appear in our dataset. Later in the period, the AP also had state-level wire services that transmitted only within a state. To the extent we are missing any content, it is most likely to be these narrowly circulated articles, e.g., state level school sports competitions that weren’t big enough news to make the front page of any preserved papers in our corpus.
>
> *Is it possible to quantify what fraction of newspaper information is captured in the Newswire database? Could the Newswire techniques also be used to capture some of these other non-wire-service domains?*
>
> Media historian Julia Guarneri estimates that by the 1910s, newswires accounted for around 50% of content in local newspapers. The methods described are readily applicable to other types of content. This said, we would urge caution as other types of content - such as syndicated opinion pieces - may be subject to copyright restrictions. We chose to focus on newswires as they are off-copyright, which allows us to make the texts publicly available. Locally written content in our corpus is off-copyright, and we agree with the reviewer that it is fascinating and pertinent to a variety of important questions. We hope that the success of Newswire will make it possible for us (or another team) to obtain the resources needed to process this local content, which is mostly contained on interior pages of the newspaper.
>
> Finally, we agree with the reviewer that historical newspapers provide an invaluable record of the past, and that similar resources may not be available for future historians studying our own period.

---

### Official Review · Reviewer_L75Y · 2024-07-18
**Review of "Newswire: A Large-Scale Structured Database of a Century of Historical News"**

**Rating:** 6
**Confidence:** 4

**Review:**

The paper is overall easy to read and understand. However, the technical approaches for processing the data lack the necessary details to understand the quality in general. The lack of details is one of the main weaknesses of the paper. This is unfortunate, as a tremendous amount of work has been put into creating this dataset, with a fairly complicated pipeline and many obstacles that had to be overcome.

The second main weakness is the fact that there is no validation of the usefulness of the dataset. While several research directions are provided regarding how the dataset could be used, there is no confirmation that this is actually possible.

Third, while many of the models mentioned in the paper are already published on Huggingface, they have no documentation and there is no model card provided.

**Strengths:**

1. Large scale dataset.
2. The dataset comes with additional enrichments such as topics and named entities.
3. The dataset could be used in a variety of use cases as well as in a variety of fields.

**Additional Feedback:**

Small points that should be clarified:
- What is the Library of Congress metadata? What does it contain?
- In the introduction, line 86, there are numbers given for all features except for topics;

**Clarity:**

The paper is easy to read and follow. However, there are many aspects that make the paper to lack self-sufficiency. There are many references to contributions by the authors or other authors that are referred to, but not sufficiently described. Many details are also provided in the appendix of the paper. However, the sections are not properly referenced in the paper, and it is unclear which details are provided, and which not.

**Correctness:**

The work looks to be correctly implemented. However, the lack of details on the training and evaluation of the variety of models that have been used makes the work difficult to assess.

**Documentation:**

The GitHub repository linked in the supplementary material is empty, no datasets or scripts can be checked.
The resources provided on Huggingface have no description, there is no model or dataset card associated with them.

**Ethics:**

There should be no ethical concerns regarding the data. The authors have discussed the necessary aspects.

**Limitations:**

The limitations of the work have been partially discussed. Limitations with regard to the methods used to process the data and extract the features have not been mentioned though.

**Opportunities For Improvement:**

In terms of clarity, some texts are repeated in the introduction, the dataset description, and the method. These parts should be rewritten, and only the details necessary for the scope of the section should be kept. Some parts of the dataset description are not particularly easy to understand. For example, it is unclear what kind of topics are tagged in the dataset, and how they were selected. The mention of classifiers is confusing here. Furthermore, it would be good to mention which types of named entities are extracted and provide a more extensive overview of these distributions at the level of the entire dataset. Sanity checks are mentioned, but it is unclear what they are, how they are applied, and what is their goal.

The choices of the topics and named entities that are extracted need to be motivated.

The processing details in lines 188-192 are not self-contained, as well as the technical details for the detection of reproduced content.

What is the SymSpell dictionary?

For most of the classifiers that are mentioned in the method section, it is unclear how they were trained, tested, or fine-tuned. It is also unclear how the manual selection of articles was done in several places of the pipeline.

In line 239 it is mentioned that 2 sets of topic tags are computed using fine-tuned neural classifiers. Which are these? Also, here it is mentioned that the classifiers were trained on 11 topics, but the statistics only mention 7 topics. Which is the correct information?

What was the performance of RoBERTa? How was the performance evaluated? The same questions apply to the S-BERT MPNet semantic similarity model.

How is the F1 score computed for the named entity recognition pipeline?

Which other off-the-shelf models were used for disambiguation? There are many details regarding training and evaluation as well as performance when describing this part of the pipeline.

**Relation To Prior Work:**

The related work section is highly focused on methods that have been borrowed or extended to meet the requirements of the dataset that is being processed. The choice of the methods from the literature is not discussed, nor thoroughly motivated. In a nutshell, it is difficult to grasp whether the chosen methods are state-of-the-art and whether more satisfactory results could have been achieved with other methods.

**Summary And Contributions:**

The paper contributes a large-scale dataset of 2.7 million unique newswires, covering the period of time from 1878 to 1977. The newswires are enriched with topics and named entities, which are subsequently linked to Wikidata.

---

> ### Author Rebuttal · Authors · 2024-08-16
>
> Thank you for the thoughtful review. We have made several changes in response to the very helpful comments, which we summarise below.
>
> *The technical approaches for processing the data lack the necessary details to understand the quality in general.*
>
> We have considerably expanded and updated the details of all the models in the appendix, including adding new sections on digitalisation and de-duplication of the content. We have also added an additional table, which is reproduced in the attached PDF, which compares each method in our pipeline against relevant alternatives. Additional information on the changes made for each stage in the pipeline is given below.
>
> *The second main weakness is the fact that there is no validation of the usefulness of the dataset. While several research directions are provided regarding how the dataset could be used, there is no confirmation that this is actually possible.*
>
> We have added a new ‘Applications’ section to the paper, to demonstrate the many uses of this dataset.
> Newswire has a diversity of uses, ranging from language model training to social science, computational linguistics, and digital humanities scholarship. Newspapers can be analogized as a first (albeit incomplete) draft of history. There are a variety of settings where including Newswire in an LLM training corpus would be useful. Not everything in that first draft ends up preserved in an online database such as Wikipedia today, and so seeing this data in training would expose an LLM to additional information. The information could also be useful as part of an external database in a retrieval augmented language modelling setup.
>
> There are other motivations for exposing an LLM to historical training data as well. For instance, Sakar and Vafa (2023) train a language model from scratch on the American Stories historical newspaper dataset to avoid "look-ahead" bias when evaluating whether an LLM can make financial predictions about the future with past data. Training on historical data can also reduce copyright risk in an uncertain legal environment.
>
> Newswire is also relevant for studying linguistic change across time, and for a diversity of questions in social science and the digital humanities. Many social scientists and historians use historical newspapers for granular information about the past. Beach and Hanlon (2023) provide a recent review. Most extant datasets can only be accessed via keyword search; indeed, Beach and Hanlon (2023) dedicate much attention to how to conduct keyword searches on historical newspapers effectively. By providing full text data, Newswire allows social scientists and historians to carry out far richer analyses of the contents of historical news. By posting the dataset on huggingface and providing tutorials, we aim to lower the costs of using Newswire in scholarly research.
>
> *Digitisation*
>
> We have added a section in the appendix which fully explains all the details on digitalisation. We also add a new table in the body of the paper which compares our chosen OCR methods against other methods. This table is reproduced in the attached PDF.
>
> We have also added a reference to the Symspell dictionary. Thank you for noticing this omission. This is a dictionary in the commonly used sense - simply a list of words in the English language.
>
> *De-duplication*
>
> We have also added an appendix section on de-duplication with all model training and evaluation details. We have similarly compared our de-duplication method to other methods in the table in the attached PDF.
>
> *Topic tagging*
>
> We have edited the section on topic tagging for clarity. We have also added an extended appendix section on topic tagging, which details how the training data were selected, how the models were trained (with all hyperparameters) and which models were used for which purposes. In the body of the paper, we have also motivated our choice of topics.
>
> We use two different sources of data to train topic tagging models. For a set of topics of particular interest during much of the century-long period that the dataset covers (Politics, Crime, Labor movement, Government regulation, Protests, Civil rights, Antitrust), we created hand-labelled, high quality training and evaluation data. We chose these topics because they cut across the period of the dataset, rather than being specific to any one time period, and they comprise large segments of news coverage.  We trained a binary topic classifier for each topic. (Selection of training data and training hyperparameters are now detailed in the appendix). We chose binary classification because we found that annotators - who were undergraduate students at our university - found it confusing to keep multiple different topic definitions top of mind at once. These are the “seven” classifiers referred to in the paper.
>
> The second type of classifier is a multi-class classifier, which categorises data into the classes from the Comparative Agendas project (27 major policy topics, such as Labor, Immigration, and Employment, Education, Environment, Energy, Immigration, Transportation). To train this, we use data from the Comparative Agendas project. They labelled 4,026 short article synopses from the New York Times according to these policy topics. As we wanted to train on articles, not on synopses, we use a semantic similarity model to match these synopses to the articles that they are summarising. We then use the retrieved articles (and the matched labels of their summaries) to train a multi-class classifier. For four topics in Comparative Agendas (sport, fires, weather, and death notices), there was minimal training data from the Comparative Agendas project, so we trained accurate binary classifiers for these four topics, which are used to replace the topic tags from this multi-class classifier. These four classifiers, plus the seven binary classifiers above, make up the full eleven classifiers.

---

> ### Author Rebuttal · Authors · 2024-08-16
>
> [Continued ...]
>
> *Sanity checks are mentioned, but it is unclear what they are, how they are applied, and what is their goal.*
>
> Thank you for pointing out that our description was unclear. We meant that when visually inspecting the plotted topic tags, they correspond with broadly known historical facts. For instance, crime coverage is elevated during Prohibition, protests and civil rights coverage peak during the 1960s, and there are spikes in defense coverage during World War I, World War II, the Korean War, and the Vietnam War. The goal was simply to indicate that there are no glaring red flags when topics are plotted.
>
>
> *NER*
>
> We tag all people, locations, organisations, and other miscellaneous entities. We select these entity types following standard practice in the NER literature. These are, for example, the same entity types that are labelled in CoNLL-2003. We have added this motivation to the paper. Figure 6 shows the distribution of these entities across the entire dataset. The overall F1 presented in the text is for a binary classification of entity/not. We also provide F1s for each entity type individually in the appendix. These are for the binary classification of each entity, so there is no separation between micro and macro F1.
>
>
> *Entity disambiguation*
>
> We have compared our results on entity disambiguation to three other entity disambiguation models from the literature. The table with these comparisons is reproduced in the attached PDF. We have further expanded the details on training and evaluation of the entity disambiguation model, both in the body of the paper and the appendix. For example, we have added a graph showing the sensitivity of entity disambiguation results to the choice of no-match threshold. This graph is reproduced in the attached PDF.
>
>
> *The related work section is highly focused on methods that have been borrowed or extended to meet the requirements of the dataset that is being processed. The choice of the methods from the literature is not discussed, nor thoroughly motivated. In a nutshell, it is difficult to grasp whether the chosen methods are state-of-the-art and whether more satisfactory results could have been achieved with other methods.*
>
> We have added a table which compares all the methods used in creating Newswire to other models in the literature. This is reproduced in the attached PDF. In general, our methods are state-of-the-art. However, there are a couple of exceptions. First, for OCR much of the data were transcribed with a highly accurate open-source OCR architecture, customised for this application. However, we started digitising newspapers at scale in 2020, using a series of small grants of credits from Microsoft Azure that we have applied for across time. The first batches were digitised with Tesseract, before the more state-of-the-art OCR architecture was developed. We don’t have the resources to re-do these batches with the more accurate EffOCR. However, it is at this point a moot issue, as we recently received credits that will allow us to use GPT for OCR error correction. It is extremely accurate and currently running. We will add a field to the dataset with OCR error corrected texts once complete. Second, de-duplication could be made slightly more accurate by using a cross-encoder in combination with the bi-encoder approach that we deploy. We elected not to do this as the difference in accuracy was minimal, and it would have added substantial additional processing cost. This is all documented in the methods comparison table in the attached PDF.
>
> *Some texts are repeated in the introduction, the dataset description, and the method. These parts should be rewritten, and only the details necessary for the scope of the section should be kept.*
>
> We have edited the paper for clarity and to limit redundancy.
>
> *The GitHub repository linked in the supplementary material is empty, no datasets or scripts can be checked. The resources provided on Huggingface have no description, there is no model or dataset card associated with them.*
>
> Thank you for flagging this oversight. The github repository and huggingface model cards are now fully updated and public.
>
> *What is the Library of Congress metadata? What does it contain?*
>
> Library of Congress metadata provides information on the location of each newspaper, dates of publication, and merges/splits of newspapers over time. We have added this information to the paper. Thank you for pointing out this omission.
>
> *In the introduction, line 86, there are numbers given for all features except for topics;*
>
> We have added that there are labels for 34 topics, for each article.

---

> > ### Comment · Reviewer_L75Y · 2024-08-26
> >
> > I would like to thank the authors for their thorough rebuttal and the way they clarified the issues raised in the review. The addition of the details mentioned in the rebuttal would really increase the quality of the submission. Overall, I believe the dataset will make a good contribution.

---

### Official Review · Reviewer_kSLJ · 2024-07-24
**Interesting resource lacking some clarity**

**Rating:** 6
**Confidence:** 4
**Clarity:** The paper is quite well-written, with…

**Review:**

This paper presents an interesting new resource, compiled using many document processing and information extraction tools. A clean, structured database of news articles is certainly of interest to the natural language processing and digital humanities fields. The paper seems to be of high quality and the authors address many issues that require news media domain expertise (near duplication of content through local re-publication, non newswire text content types, etc). Throughout the paper were small analyses showing how various aspects (e.g. article topic) match historical events over a century-long time period - I found this particularly interesting. However, on careful examination, I find it difficult to determine the novel contributions of this paper. Many of the most impressive components seem to be direct applications of tools developed by groups of the same authors. Rather than providing details of those methods or deeper analysis, we are entirely referred to existing work. For this reason, I find the paper technically interesting, while being a somewhat incremental application of existing methods and datasets. I expand on these strengths and weaknesses below and hope that the authors will be able to clarify.

**Strengths:**

- The resource is interesting and required careful expertise to create (e.g. domain specific knowledge).
- Extracting text from many diverse images is challenging and the authors use sophisticated pipelines to perform this.
- The presented dataset is certain to be impactful in fields like digital humanities.

**Additional Feedback:**

None

**Correctness:**

Aside from issues raised in the “Improvements” section, this paper appears technically sound.

**Documentation:**

The dataset is provided on huggingface. Details are provided regarding the license. **However, the provided code repository is empty.** Having access to the code would help clarify which parts of this paper were created by prior work and which parts represent novel contributions.

**Limitations:**

The limitations section is well-written and very informative. The authors were careful to justify why these news articles can be permissively used. The note on copyright protections for these types of datasets is great. The authors also make an excellent point in calling out that historical news documents may contain offensive content or inaccurate facts.

**Opportunities For Improvement:**

This paper is an extremely interesting resource and I enjoyed reading it. However, several components seem to be directly taken from prior work and the new components are not explained with sufficient detail.

### Relationship to Prior Work

Some relationships to prior work are unclear, as this paper appears to directly re-use pipelines from closely related datasets, without providing any additional technical details or changes.

Below are the components and my concerns about each.
- Collecting the dataset and using OCR to extract text from a difficult format (“Digitization”)
  - Multiple parts of the paper refer to the “American Stories” resource as the origin of this pipeline. That paper, published in NeurIPS 2023 Datasets and Benchmarks, appears to be a collection of OCRd newspaper articles. **Are the text articles in your Newswire dataset part of this resource?**
  - The paper states “The layout analysis… is documented and evaluated in more detail in (4)” and “These steps are also described in more detail in (4)”, where 4 is the American Stories paper. There are no further details or analysis for the OCR component.
- Finding and removing near duplicates (“Detection of reproduced content”)
  - Similarly, this section briefly describes a duplicate detection pipeline and states “All technical details are provided in (14)”, where 14 is an ICLR 2023 publication.
- Finding locations (“Georeferencing”)
  - This section describes an n-gram matching protocol against GeoNames locations. Accuracy against hand labels are stated, with no further analysis or baselines.
- Topic tagging
  - This section describes new components, but it is unclear to me why both individual binary classifiers and semantic similarity retrieval models were used, and which produce the final labels.
- Named entity recognition.
  - This section very briefly describes a new NER model and does not contain sufficient detail. It refers to the appendix, but the appendix section is simply a results table.
- Entity disambiguation.
  - This section makes novel contributions, but most of the details are pushed to the supplementary materials. I think focusing on the contributions here, and showing metrics that justify your design decisions, would improve the paper.

To summarize: several key components of this work seem to draw entirely on other publications (e.g. “All technical details are provided in (14)”). The new components are not described in sufficient detail - missing details on hand-annotation, baselines, and justifications for design choices. Moving some details from the appendix to the main text could help.

### Clarifying other Details

Many model accuracy statistics are given without comparisons, baselines, or values. Some data is human annotated, but no inter-annotator agreement statistics are provided. For example, there is no baseline for NER/Appendix 2.3 and 284-289 could provide a ROC curve for tradeoffs at various thresholds. 216-221 describes a filtering process but does not give context for “nearly 96% accuracy”.

### Framing Issues

While I agree that historical documents are a very interesting resource to study, the first line of the paper incorrectly suggests that this resource could be useful “As contemporary data sources for large language model training become depleted”. This resource is 2.7M documents and appears to be 10s of GB including structured information. A typical LLM pretraining dataset is in the billions of documents and 1-10TB range - *it is not clear that those data sources are depleted.*

Aside from scale, justifying this statement would require that the knowledge found in these articles cannot be found elsewhere [in LLM data]. I believe the authors do the opposite, as they successfully link many article subjects to Wikipedia/Wikidata. Perhaps future coverage analysis could highlight specific instances in this dataset that would be otherwise missing.

### References:
[4] [American Stories: A Large-Scale Structured Text Dataset of Historical U.S. Newspapers](https://neurips.cc/virtual/2023/poster/73515)
Melissa Dell, Jacob Carlson, Tom Bryan, Emily Silcock, Abhishek Arora, Zejiang Shen, Luca D'Amico-Wong, Quan Le, Pablo Querubin, Leander Heldring

[14] [Noise-Robust De-Duplication at Scale](https://openreview.net/forum?id=bAz2DBS35i)
Emily Silcock, Luca D'Amico-Wong, Jinglin Yang, Melissa Dell

**Relation To Prior Work:**

This is my main area of concern, see section under Opportunities for Improvement.

**Summary And Contributions:**

This paper presents *Newswire*, a dataset compiled from OCRd U.S. news articles from 1878 to 1977. This involves obtaining text through a custom OCR pipeline, finding near duplicate articles, and tagging with information extraction approaches (geolocation, topics, entity recognition and linking with Wikidata). Each of those aspects requires a customized pipeline component, such as encoding models tuned to find similar articles or an entity disambiguation system. The authors also note that articles from this time period are free from copyright restrictions, and appear to be gathered from the US Library of Congress.

---

> ### Author Rebuttal · Authors · 2024-08-16
>
> Thank you for your thoughtful comments. Below, we provide a brief response to your questions and comments.
>
> *Multiple parts of the paper refer to the “American Stories” resource as the origin of this pipeline. That paper, published in NeurIPS 2023 Datasets and Benchmarks, appears to be a collection of OCRd newspaper articles. Are the text articles in your Newswire dataset part of this resource?*
>
> Only 15% of the articles in Newswire come from American Stories. The rest are from other sources and have not previously been published. The content in American Stories is overwhelmingly from before 1920, whereas Newswire extends through 1977. American Stories does not detect reproduced content, tag topics or entities, or disambiguate entities.
>
>
> *Digitisation - The paper states “The layout analysis… is documented and evaluated in more detail in (4)” and “These steps are also described in more detail in (4)”, where 4 is the American Stories paper. There are no further details or analysis for the OCR component.*
>
> We have added an appendix with full details about the methods used for layout detection and OCR.
>
> *Finding and removing near duplicates (“Detection of reproduced content”) - Similarly, this section briefly describes a duplicate detection pipeline and states “All technical details are provided in (14)”, where 14 is an ICLR 2023 publication.*
>
> Similarly, we have added an appendix with full details of the deduplication pipeline.
>
> *Filtering to newswire content - 216-221 describes a filtering process but does not give context for “nearly 96% accuracy”.*
>
> The accuracy metric reported is based on the performance of our classifier on a hand-labelled test set of 448 samples. More details regarding the training are available in the appendix, and we have added clarification in the main text.
>
> *Finding locations (“Georeferencing”) - This section describes an n-gram matching protocol against GeoNames locations. Accuracy against hand labels are stated, with no further analysis or baselines.*
>
> More details of this method are given in the appendix. We have more clearly signposted this in the body of the paper. Additionally we have added a new table comparing all methods used to relevant baselines. This is reproduced in the attached PDF.
>
> *Topic tagging - this section describes new components, but it is unclear to me why both individual binary classifiers and semantic similarity retrieval models were used, and which produce the final labels.*
>
> We have clarified the text explaining the topic tagging in the paper.
> We have two types of classifiers. The first is for topics of particular interest across a substantial share of the century-long period covered by the dataset (Politics, Crime, Labor movements, Government regulation, Protests, Civil rights, and Antitrust). For these we created hand-labelled, double-annotated training and evaluation data, and trained a binary topic classifier for each topic. We chose binary topic classifiers because we found in practice that our annotators (undergraduate students at our university) had a hard time keeping multiple topic definitions in mind simultaneously.
>
> The second type of classifier is a multi-class classifier, which categorises data into the classes from the Comparative Agendas project (30 major policy topics, such as Labor, Immigration, and Employment, Education, Environment, Energy, Immigration, Transportation). To train this, we use data from the Comparative Agenda project, as they had labelled 4,026 short article synopses from the New York Times according to these policy topics. As we wanted to train on articles, not on synopses, we use a semantic similarity model to match these synopses to the articles that they are summarising. We then use the retrieved articles (and the matched labels of their summaries) to train a multi-class classifier. Hence the semantic similarity model is only used for creating the training data, since Comparative Agendas labelled article synopses and did not provide information on which full length articles these synopses referred to.
>
> *Named entity recognition - This section very briefly describes a new NER model and does not contain sufficient detail. It refers to the appendix, but the appendix section is simply a results table.*
>
> We have added additional information on the NER model to the appendix. We have also compared the NER model to relevant baselines. These results are reported in the table in the attached PDF, which will be added to the paper.
>
> *Entity disambiguation - This section makes novel contributions, but most of the details are pushed to the supplementary materials. I think focusing on the contributions here, and showing metrics that justify your design decisions, would improve the paper.*
>
> We have moved some additional details of the entity disambiguation model to the main body of the paper. We have also added a graph showing how the accuracy of the model varies at different thresholds, which is reproduced in the attached PDF. Additionally, we have added comparisons of the chosen model to relevant baselines. The results of this are also in the attached PDF.
>
>
> *While I agree that historical documents are a very interesting resource to study, the first line of the paper incorrectly suggests that this resource could be useful “As contemporary data sources for large language model training become depleted”. This resource is 2.7M documents and appears to be 10s of GB including structured information. A typical LLM pretraining dataset is in the billions of documents and 1-10TB range - it is not clear that those data sources are depleted.*
>
> We believe this dataset provides a blueprint for how historical and off-copyright text data can be unlocked for use with LLMs. Nonetheless, we agree that the dataset is small compared to LLM pretraining datasets, and we have modified the framing in this sentence.

---

> > ### Comment · Reviewer_kSLJ · 2024-09-01
> > **Thanks to the authors for excellent clarifications**
> >
> > My apologies for the delayed reply. I very much appreciate the clarifications made by the authors in their rebuttal and the revised text. I see that some of my concerns were misplaced or alleviated. The information in the appendix is very welcome - consider moving some of that content to the main text, space permitting. I also appreciate the specific examples of entities that are NOT present in Wikipedia - very interesting. Thanks for these additions!
> >
> > I will raise my score by 1 point (final rating: 6). I can't seem to edit my original entry in openreview, but I will leave a note in the committee member chat to ensure the increased score is taken into consideration.

---

> ### Author Rebuttal · Authors · 2024-08-16
>
> [Continued ...]
>
> *Aside from scale, justifying this statement would require that the knowledge found in these articles cannot be found elsewhere [in LLM data]. I believe the authors do the opposite, as they successfully link many article subjects to Wikipedia/Wikidata. Perhaps future coverage analysis could highlight specific instances in this dataset that would be otherwise missing.*
>
> While we link many individuals to Wikipedia, this dataset also allows us to explore which individuals were commonly-mentioned at the time, but not remembered in sources such as Wikipedia. For example, we find that John Lewis Niblack (journalist and judge in Indiana who exposed the KKK), and Arnold Maremont (industrialist and the head of the Illinois Public Aid Commission) were frequently mentioned entities in the 1960s, yet lack a Wikipedia page.
>
> *Github repository*
> Thank you for flagging this oversight. The github repository is now fully updated and public.

---

### Official Review · Reviewer_L8QG · 2024-07-24
**Newswire: A Large-Scale Structured Database of a Century of Historical News**

**Rating:** 8
**Confidence:** 3
**Correctness:** The claims are correct.
**Clarity:** The paper is well-organized and well-…

**Review:**

### **Quality**
The evaluation metrics indicate high precision and recall for OCR, de-duplication, and named entity recognition, demonstrating the robustness of the methodologies used. However, it could benefit from a more detailed error analysis to discuss common OCR errors and de-duplication challenges and the paper lacks detailed metrics on the scalability and computational efficiency of the processing pipeline. The paper lacks detailed ablation studies to show the impact of each component (e.g., OCR, de-duplication) on the overall dataset quality. Also, more metrics on the scalability and computational efficiency of the processing pipeline would strengthen the methodology. Finally, error analysis discussing common OCR errors, challenges in de-duplication, and entity recognition limitations would provide a fuller picture of the methodology's effectiveness.

### **Clarity**
The paper is well-organized, with clear sections on introduction, related work, dataset construction, methodologies, evaluations, and limitations. The writing is clear and concise, making the paper accessible to a broad audience. The paper could benefit from additional visual aids, such as flowcharts or diagrams, to illustrate the processing pipeline and methodology steps more clearly. Additionally, while potential applications are mentioned, specific use cases and examples of research applications would enhance clarity.

### **Originality**
The creation of a century-spanning dataset of U.S. news wire articles is a novel contribution, filling an important gap in historical news datasets. The use of advanced neural models for de-duplication, georeferencing, topic tagging, and entity disambiguation demonstrates originality in methodology. The paper could strengthen its originality claims by providing more extensive comparisons with other historical news datasets and benchmarks.

### **Significance**
The dataset has the potential to impact various fields, including NLP, social sciences, and digital humanities, by providing a rich resource for historical analysis and machine learning applications. Its structured nature makes it suitable for a wide range of applications, from training large language models to conducting detailed historical research. However, although the dataset's potential applications are broad, the paper could benefit from including specific examples of how the dataset has been or can be used in real-world research projects.

**Strengths:**

### Significance of the Paper
The significance of the paper has been articulated in the prior sections.

### Relevance to the Broader Research Community
The dataset is relevant to multiple fields, including NLP, computational linguistics, social sciences, and digital humanities. Its structured nature makes it ideal for training large language models and other machine-learning applications. Historians and social scientists may be able to leverage the dataset for in-depth analysis of historical trends, media coverage, and social changes over time.

### Quality of the Research
The paper employs state-of-the-art techniques for OCR, de-duplication, georeferencing, topic tagging, and named entity recognition, ensuring high-quality data processing. Detailed evaluations using precision, recall, and other metrics demonstrate the robustness and reliability of the dataset. The public availability of the dataset and processing pipelines on Hugging Face supports reproducibility and transparency in research.

### Ethical and Social Implications
The dataset is derived from public-domain news articles, ensuring that it is freely accessible and legally usable for research purposes. By digitizing and structuring historical news articles, the dataset helps preserve important historical records and makes them accessible for future generations. The paper acknowledges the potential for biases in historical texts and the need for careful analysis and interpretation. This consideration is crucial for ethical research and ensuring balanced representation in studies using the dataset.

**Additional Feedback:**

I have no additional feedback.

**Documentation:**

There is sufficient detail on data collection and organization, availability, maintenance and ethical and responsible use.

**Ethics:**

There are no ethical concerns.

**Limitations:**

The authors have adequately the work's limitations.

**Opportunities For Improvement:**

### **Significance of the Paper**
While the dataset is extensive and covers a century of historical news, it is limited to U.S. news wire articles. This geographical focus restricts its applicability for researchers interested in global historical perspectives or comparative studies involving news from multiple countries. Additionally, the dataset ends in 1977 due to copyright restrictions, which means more recent historical events are not covered, nor the evolution of news coverage for the past 45+ years.

### **Relevance to the Broader Research Community**
Although the dataset has broad applicability, its utility may be limited by the inherent biases present in historical news articles. The language and perspectives reflected in the dataset include outdated or offensive views, which may be useful in studies on social change and cultural evolution. Researchers will need to account for these biases, potentially complicating their analyses. Furthermore, while the dataset is rich in metadata, the absence of detailed comparison with other existing datasets limits its position within the broader landscape of historical text archives.

### **Quality of the Research**
The paper could benefit from a more detailed error analysis, particularly in OCR and de-duplication processes. Understanding common OCR errors and de-duplication challenges would provide a clearer picture of the dataset's limitations. Additionally, the paper lacks ablation studies that could illustrate the contribution of each component of the processing pipeline. Without these studies, it is difficult to assess the individual impact of layout recognition, OCR, topic tagging, and named entity recognition on the overall quality of the dataset.

### **Ethical and Social Implications**
While the paper acknowledges potential biases in historical texts, it does not provide detailed strategies for mitigating these biases in subsequent research. There is also a lack of discussion on how to handle potentially sensitive content within the dataset. Moreover, the focus on public domain articles, while ethical and legal, excludes a vast amount of proprietary historical news data that could provide a more comprehensive view of history.

In summary, the paper presents a high-quality dataset with broad relevance, but it is limited by its geographical scope, potential biases in the source material, and the need for more detailed error analysis and ablation studies. Addressing these limitations in future work would enhance the dataset's utility and reliability.

**Relation To Prior Work:**

The work's relation to prior work is clearly discussed.

**Summary And Contributions:**

The paper presents the Newswire dataset, a comprehensive collection of 2.7 million unique public domain U.S. news wire articles spanning from 1878 to 1977. This dataset aims to provide a rich, structured resource for historical analysis, machine learning, and various research applications. It is potentially valuable for training large language models, conducting computational linguistics research, performing social science studies, and exploring digital humanities.

The dataset includes a century of news articles, making it one of the most extensive historical news datasets available. Articles are georeferenced, tagged with topics, and recognized for named entities, and disambiguated individuals.

The paper presents a processing pipeline that uses state-of-the-art methods to recognize newspaper layouts and transcribe text from raw image scans. It employs a customized neural bi-encoder model to identify and remove duplicated articles, ensuring uniqueness. It uses fine-tuned neural classifiers to tag topics and geo-reference locations within articles and it tags and disambiguates named entities to Wikipedia, enhancing the dataset's usability.

---

> ### Author Rebuttal · Authors · 2024-08-17
>
> Many thanks for the thoughtful review. We are glad that you found the paper valuable. Below, we provide a brief response to your very helpful questions and comments.
>
> *Error analysis discussing common OCR errors, challenges in de-duplication, and entity recognition limitations would provide a fuller picture of the methodology's effectiveness.*
>
> We have added an error analysis section for each of these methods to the appendix.
>
> *More metrics on the scalability and computational efficiency of the processing pipeline would strengthen the methodology*
>
> We have added extra discussion of the computational efficiency of each of our chosen methods. Concerns around computational efficiency were central in our design of the pipeline, as we aimed to develop an approach that could be applied to large-scale newspaper collections on a highly constrained academic budget.
>
> *The paper lacks detailed ablation studies to show the impact of each component (e.g., OCR, de-duplication) on the overall dataset quality*
>
> We have added a new table comparing all methods used to relevant baselines. This is reproduced in the attached PDF.
>
> *The paper could benefit from additional visual aids, such as flowcharts or diagrams, to illustrate the processing pipeline and methodology steps more clearly.*
>
> We agree with the reviewer that additional visual aids would be useful and will add this.
>
> *While potential applications are mentioned, specific use cases and examples of research applications would enhance clarity.*
>
> We have added a new ‘Applications’ section to the paper, to demonstrate the many uses of this dataset. Newswire has a diversity of uses, ranging from language model training to social science, computational linguistics, and digital humanities scholarship. Newspapers can be analogized as a first (albeit incomplete) draft of history. There are a variety of settings where including Newswire in an LLM training corpus would be useful. Not everything in that first draft ends up preserved in an online database such as Wikipedia today, and so seeing this data in training would expose an LLM to additional information. The information could also be useful as part of an external database in a retrieval augmented language modeling setup.
>
> There are other motivations for exposing an LLM to historical training data as well. For instance, Sakar and Vafa (2023) train a language model from scratch on the American Stories historical newspaper dataset to avoid "look-ahead" bias when evaluating whether an LLM can make financial predictions about the future with past data. Training on historical data can also reduce copyright risk in an uncertain legal environment.
>
> Newswire is also relevant for studying linguistic change across time, and for a diversity of questions in social science and the digital humanities. Many social scientists and historians use historical newspapers for granular information about the past. Beach and Hanlon (2023) provide a recent review. Most extant datasets can only be accessed via keyword search; indeed, Beach and Hanlon (2023) dedicate much attention to how to conduct keyword searches on historical newspapers. By providing full text data, Newswire allows social scientists and historians to carry out far richer analyses of the contents of historical news. By posting the dataset on huggingface and providing tutorials, we aim to lower the costs of using Newswire in scholarly research.
>
> *The paper could strengthen its originality claims by providing more extensive comparisons with other historical news datasets and benchmarks.*
>
> We have expanded the discussion of related literature to have a more extended discussion of other historical news datasets.
>
> *While the dataset is extensive and covers a century of historical news, it is limited to U.S. news wire articles. This geographical focus restricts its applicability for researchers interested in global historical perspectives or comparative studies involving news from multiple countries. Additionally, the dataset ends in 1977 due to copyright restrictions, which means more recent historical events are not covered, nor the evolution of news coverage for the past 45+ years.*
>
> While the dataset is limited to the U.S. we hope that the methodology, models and training data will be useful for researchers in other contexts. Our training data, for example, could be machine translated to train models for other languages, and the newspaper layout analysis model provides a good start for recognizing newspaper layouts even when the language is different. Copyright law restrictions are unfortunate, leaving a gap between 1978 and the more recent period, when news is preserved by publishers through digital channels. While output cannot be shared publicly, academic researchers could apply our models to copyrighted content and use the output for non-commercial, academic ends under fair use, if desired. Our publicly available pipeline would dramatically lower the cost of doing this.

---

> ### Author Rebuttal · Authors · 2024-08-17
>
> *Although the dataset has broad applicability, its utility may be limited by the inherent biases present in historical news articles. The language and perspectives reflected in the dataset include outdated or offensive views, which may be useful in studies on social change and cultural evolution. Researchers will need to account for these biases, potentially complicating their analyses. While the paper acknowledges potential biases in historical texts, it does not provide detailed strategies for mitigating these biases in subsequent research. There is also a lack of discussion on how to handle potentially sensitive content within the dataset.*
>
> While the dataset contains an array of biases, in general we see this as a feature rather than a bug. Keeping all articles in the dataset is important for developing a more comprehensive, less selected view of history. Claude, for example, refuses to process many articles about World War I, labelling them as harmful due to the discussion of violence. If we had made such a judgement, this would significantly bias our representation of U.S. news during that period. Any sort of filtering on our part would make the dataset largely useless for social science research, by exacerbating selection bias into what was included. Instead, we leave it up to downstream users of Newswire to choose how to filter the dataset.
>
> At some level, many people (especially in the digital humanities) would subscribe to the view that all content is biased. When you read coverage of the same historical event, it is almost always the case that each different article describing the event chooses to include a different subset of facts pertaining to the event. In reading a different set of facts, the reader will often come to quite distinct conclusions about what happened and what this implies about the world. Which is the “unbiased” version, given any article will have to simplify what it reports about a complex reality? Of course, people don’t agree on that, and the Associated Press often would provide several different articles describing the same event that include a different subset of the facts around it. Studying this is one of the things that makes the dataset really fascinating.
>
> We do of course appreciate that some elements will need to be removed for some purposes, but the broader point is that what is meant by bias may differ dramatically depending on the application. Because there are many potential applications of Newswire, it is difficult to give meaningful general purpose guidance. Instead, we leave the judgement on what should be removed to the user.
>
> *The focus on public domain articles, while ethical and legal, excludes a vast amount of proprietary historical news data that could provide a more comprehensive view of history.*
>
> Many proprietary news sources maintain their own archives, or have archives managed by third party sources. For example, the New York Times can be found on ProQuest historical newspapers from the 1880s, as can the Wall Street Journal. Newswire is constructed from the content published in tens of thousands of local newspapers, mostly from smaller towns across America. Hence, it can help to increase the representativeness of historical newspaper data, which to date has primarily focused on the largest newspapers.

---

### Decision · Program_Chairs · 2024-09-26

**Decision:**

Accept (Poster)

**Comment:**

The proposed dataset is very relevant to the community, well-documented and well-described in the paper. It can be of use above and beyond the ML community as well, by historians and members of the STS community. There are minor improvements to be made, as proposed by the reviewers, and it would be ideal to make them in the camera-ready version of the submission.